# High-throughput, microscope-based sorting to dissect cellular heterogeneity

Nicholas Hasle[1], Anthony Cooke[2], Sanjay Srivatsan[1], Heather Huang[3] (iD), Jason J Stephany[1],
Zachary Krieger[1], Dana Jackson[1], Weiliang Tang[4], Sriram Pendyala[1], Raymond J Monnat Jr.[1,4],
Cole Trapnell[1], Emily M Hatch[3] & Douglas M Fowler[1,5,*] (iD)

## Abstract

Microscopy is a powerful tool for characterizing complex cellular phenotypes, but linking these phenotypes to genotype or RNA expression at scale remains challenging. Here, we present Visual Cell Sorting, a method that physically separates hundreds of thousands of live cells based on their visual phenotype. Automated imaging and phenotypic analysis directs selective illumination of Dendra2, a photoconvertible fluorescent protein expressed in live cells; these photoactivated cells are then isolated using fluorescence-activated cell sorting. First, we use Visual Cell Sorting to assess hundreds of nuclear localization sequence variants in a pooled format, identifying variants that improve nuclear localization and enabling annotation of nuclear localization sequences in thousands of human proteins. Second, we recover cells that retain normal nuclear morphologies after paclitaxel treatment, and then derive their single-cell transcriptomes to identify pathways associated with paclitaxel resistance in cancers. Unlike alternative methods, Visual Cell Sorting depends on inexpensive reagents and commercially available hardware. As such, it can be readily deployed to uncover the relationships between visual cellular phenotypes and internal states, including genotypes and gene expression programs.

**Keywords** genetic screening; microscopy; pharmacology; subcellular localization; transcriptomics

**Subject Category** Methods & Resources

**Mol Syst Biol. (2020) 16: e9442**

## Introduction

High content imaging (Boutros *et al*, 2015), *in situ* sequencing methods (Lee *et al*, 2014, 2015; Chen *et al*, 2015; Moffitt *et al*, 2016;

Emanuel *et al*, 2017; Eng *et al*, 2019; Feldman *et al*, 2019; Wang *et al*, 2019), and other approaches (Chien *et al*, 2015; Binan *et al*, 2016, 2019; Kuo *et al*, 2016; David *et al*, 2017) have revolutionized the investigation of how genetic variants and gene expression programs dictate cellular morphology, organization, and behavior. One important application of these methods is visual genetic screening, in which a library of genetic variants is introduced into cells and the effect of each variant on a visual phenotype is quantified. In a classical high content visual genetic screen, each genetic perturbation occupies a separate well. New *in situ* methods, which employ sequencing by repeated hybridization of fluorescent oligo probes (Chen *et al*, 2015; Moffitt *et al*, 2016; Emanuel *et al*, 2017; Eng *et al*, 2019; Wang *et al*, 2019) or direct synthesis (Ke *et al*, 2013; Lee *et al*, 2014, 2015; Feldman *et al*, 2019) to visually read out nucleic acid barcodes, permit hundreds of perturbations to be assessed in a pooled format. For example, multiplexed fluorescent *in situ* hybridization was used to assess the effect of 210 CRISPR sgRNAs on RNA localization in ~30,000 cultured human U-2 OS cells (Wang *et al*, 2019), and *in situ* sequencing was used to measure the effect of 963 gene knockouts on the localization of an NFkB reporter at a throughput of ~3 million cells (Feldman *et al*, 2019). Visual phenotyping methods can also dissect non-genetic drivers of phenotypic heterogeneity. Here, characterization of cells with distinct visual phenotypes can reveal different cell states—such as signaling pathway activities and gene expression profiles—that are associated with different cellular morphologies. For example, the photoactivatable marker technology single-cell magneto-optical capture was used to isolate cells that successfully resolved ionizing radiation-induced DNA damage foci (Binan *et al*, 2019).

Despite their utility, current methods have limitations (Table EV1). Some, such as high content imaging, require highly specialized or custom-built hardware. Others, like *in situ* sequencing, employ complex protocols, sophisticated computational pipelines, and expensive dye-based reagents. Methods that mark and sort for individual cells with a photoactivatable protein or compound are simpler and less expensive. However, they are either low

---

1  Department of Genome Sciences, University of Washington, Seattle, WA, USA
2  Leica Microsystems, Buffalo Grove, IL, USA
3  Divisions of Basic Sciences and Human Biology, Fred Hutchinson Cancer Research Center, Seattle, WA, USA
4  Department of Pathology, University of Washington, Seattle, WA, USA
5  Department of Bioengineering, University of Washington, Seattle, WA, USA
   *Corresponding author. Tel: +1 206 221 5711; E-mail: dfowler@uw.edu

throughput (< 1,000 cells per experiment; Chien *et al*, 2015; Binan *et al*, 2016, 2019; Kuo *et al*, 2016) or lack single-cell specificity (David *et al*, 2017). Furthermore, they cannot investigate more than one or two phenotypes per experiment.

To address these shortcomings, we developed Visual Cell Sorting, a flexible and simple high-throughput method that uses commercial hardware to enable the investigation of cells according to visual phenotype. Visual Cell Sorting is an automated platform that directs a digital micromirror device to mark single live cells that express a nuclear photoactivatable fluorescent protein for subsequent physical separation by fluorescence-activated cell sorting (FACS). We demonstrate that Visual Cell Sorting enables visual phenotypic sorting into 4 bins, increases the throughput of cellular separation by 1,000-fold compared to other single-cell photoconversion-based technologies (Chien *et al*, 2015; Binan *et al*, 2016, 2019; Kuo *et al*, 2016), and permits pooled genetic screening and transcriptomic profiling. For example, Visual Cell Sorting enabled us to sort hundreds of thousands of cultured human cells according to the nuclear localization of a fluorescent reporter protein and thus score a library of nuclear localization sequence variants for function. In a second application, we isolated paclitaxel-treated cells with normal or lobulated nuclear morphologies and subjected each population to single-cell RNA sequencing, revealing multiple pathways associated with paclitaxel resistance. Visual Cell Sorting requires simple, inexpensive, and commercially available wide-field microscope hardware, routine genetic engineering, and a standard 4-laser FACS instrument to perform. As such, we envision that Visual Cell Sorting can readily be deployed to uncover the relationships between visual cellular phenotypes and their associated internal states, including genotype and gene expression programs.

## Results

### Physical separation of cells by visual phenotype

Visual Cell Sorting uses FACS to separate hundreds of thousands of cells by their visual phenotypes. Cells are first modified to express Dendra2, a green-to-red photoconvertible fluorescent protein (Chudakov *et al*, 2007) that will act as a phenotypic marker and enable downstream FACS sorting. Next, cells are imaged on an automated microscope. In each field of view, cells are identified and analyzed for phenotypes of interest. According to their phenotype, cells are illuminated with 405 nm light for different lengths of time using a digital micromirror device, resulting in different levels of red Dendra2 fluorescence (Fig EV1A). The imaging, analysis, and photoactivation steps are performed at each field of view, and unlike previous photoactivatable marker-based methods, these steps are automated, allowing hundreds of thousands of cells to be assessed per experiment. Once all cells have been imaged, analyzed, and photoactivated, FACS is used to sort them into bins according to their level of Dendra2 photoactivation (Fig 1A).

We first sought to establish the single-cell accuracy of Dendra2 photoactivation, and whether variable photoactivation states could be discerned by flow cytometry. We noticed that similar technologies use photoactivatable dyes or proteins localized to the whole cell body (Chien *et al*, 2015; Binan *et al*, 2016, 2019; Kuo *et al*, 2016). This localization strategy makes identifying the boundaries of the

fluorescent signal difficult, which results in partial photoactivation or photoactivation of the marker in a cell adjacent to a cell of interest. With this in mind, we expressed Dendra2 in the nucleus either as a histone H3 fusion (H3-Dendra2) or with an upstream nuclear localization sequence (NLS-Dendra2x3). The boundaries of nuclear Dendra2 signal are easy to identify, permitting quantitative photoactivation of Dendra2 in the cells of interest, and the cytoplasm provides a spacer between the Dendra2 in different cells, reducing photoactivation of cells adjacent to the cells of interest.

To measure photoactivation accuracy, H3-Dendra2-positive cells co-expressing H2B-miRFP (Shcherbakova *et al*, 2016) were mixed with cells expressing H3-Dendra2 alone at decreasing ratios. We instructed the microscope to activate Dendra2 in cells harboring miRFP-positive nuclei, and then, we quantified the co-occurrence of miRFP and activated Dendra2 florescence signals using flow cytometry (Fig 1B). The ratio of activated Dendra2 fluorescence to unactivated Dendra2 fluorescence (Dendra2 photoactivation ratio) accurately predicted whether a cell was miRFP-positive, even when the miRFP-expressing cells were present at ~0.5% frequency, with average precision of 94% and recall of 80% (Fig 1C).

Previous photoactivatable marker-based methods have been limited to two photoactivation levels: activated and unactivated. To test whether we could encode more than one photoactivation level, and thus more than one phenotype, we exposed different cells in the same well to 405 nm light for 0, 50, 200, or 800 ms. Flow cytometry of the Dendra2 fluorescence distribution showed four distinct levels of Dendra2 photoactivation, indicating that Visual Cell Sorting can sort four different visual phenotypes or four discrete bins of a continuous phenotype (Fig 1D). Furthermore, these four photoactivation levels can still be distinguished over 12 h following activation (Fig EV1B, left panel). To extend the amount of time that the photoactivation levels remain distinct from one another, we placed H3-Dendra2 expression under the control of a doxycycline-inducible promoter. By shutting off Dendra2 expression before the experiment, the 50-, 200-, and 800-ms photoactivation levels remained distinguishable for up to 24 h (Fig EV1B, right panel). Finally, we examined the effect of Dendra2 photoactivation on cell viability and function. Activated cells did not exhibit higher rates of apoptosis or cell death even 2 days after photoactivation, nor did we detect effects of photoactivation on gene expression (Fig EV1C and D). These results indicate that Dendra2 photoactivation does not appreciably affect cell survival or gene expression programs.

### Visual cell sorting enables pooled, image-based genetic screening

To test whether Visual Cell Sorting enables image-based genetic screening, we asked if we could separate cells according to the nuclear localization of a fluorescent reporter protein. Nuclear localization sequences (NLSs) are short peptides that direct proteins to the nucleus, and NLSs are critical for the function of thousands of human transcription factors, nuclear structural proteins, and chromatin modifying enzymes. Over 90% of nuclear proteins do not have an annotated nuclear localization sequence in UniProt, and current NLS prediction algorithms cannot sensitively identify known NLSs without drastically decreasing their precision (Nguyen Ba *et al*, 2009; Lin & Hu, 2013). This shortcoming may arise because these NLS prediction algorithms rely on sequence alignments or amino acid frequencies of naturally observed NLSs, which are subject to discovery bias.

Therefore, we used Visual Cell Sorting to evaluate a large library of NLS missense variants; sort cells according to the NLS function; and sequence the sorted cells (Fig 2A), with the hypothesis that the resulting data could be used to improve NLS prediction.

We based our library on the SV40 NLS, a 7-residue sequence containing a lysine and arginine-rich region (K/R motif) that was

the first NLS to be discovered (Kalderon *et al*, 1984). To assess NLS variant function, we constructed a fluorescent nuclear localization reporter similar to one described previously (Kalderon *et al*, 1984). Cultured U-2 OS H3-Dendra2 cells expressing the wild-type SV40 NLS fused to a CMPK-miRFP reporter had high levels of miRFP in the nucleus, relative to the cytoplasm. The degree of nuclear

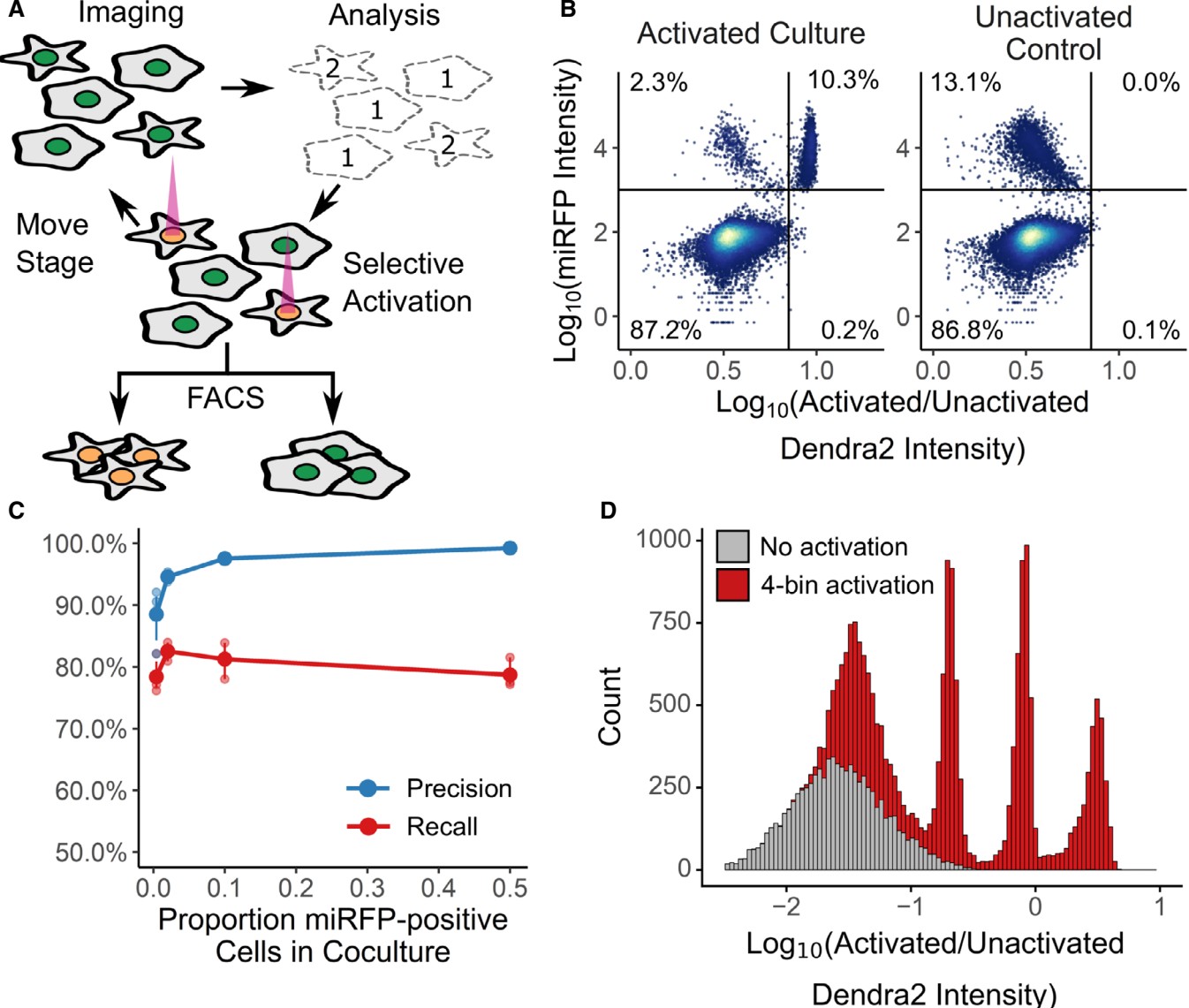

**Figure 1. Visual Cell Sorting.**

A    In an automated fashion, cells in a field of view are imaged and their phenotype classified. Cells of interest are illuminated with 405 nm light, which irreversibly photoactivates Dendra2 from its green to its red fluorescent state. The microscope then moves to a new field of view. These steps are repeated across an entire culture well. Then, fluorescence-activated cell sorting based on Dendra2 photoactivation is used to physically recover cells of interest.

B    To assess the photoactivation accuracy, U-2 OS cells expressing nuclear Dendra2 and miRFP, or nuclear Dendra2 alone, were co-cultured. The microscope was programmed to activate Dendra2 in cells expressing miRFP. Following photoactivation, miRFP expression and the ratio of activated to unactivated Dendra2 (left panel, *n* = 18,766 cells) were assessed with flow cytometry. In a second co-culture, Dendra2 was unactivated (right panel, *n* = 18,395 cells). Lines indicate gates for miRFP-positive cells and activated Dendra2 cells, with the percentage of cells appearing in each quadrant indicated.

C    Same experiment as (B), except cells were mixed such that 0.5%, 4%, 12%, or 50% were miRFP positive. Precision and recall were computed; large solid points, mean (*n* = 3 replicates); small points, individual replicate values; error bars, standard error from the mean.

D    U-2 OS cells in one well were illuminated with 405 nm light for 0, 50, 200, or 800 ms (red; *n* = 16,397). Cells in a second well were left unactivated (gray; *n* = 8,497). The ratio of activated to unactivated Dendra2 was determined by flow cytometry.

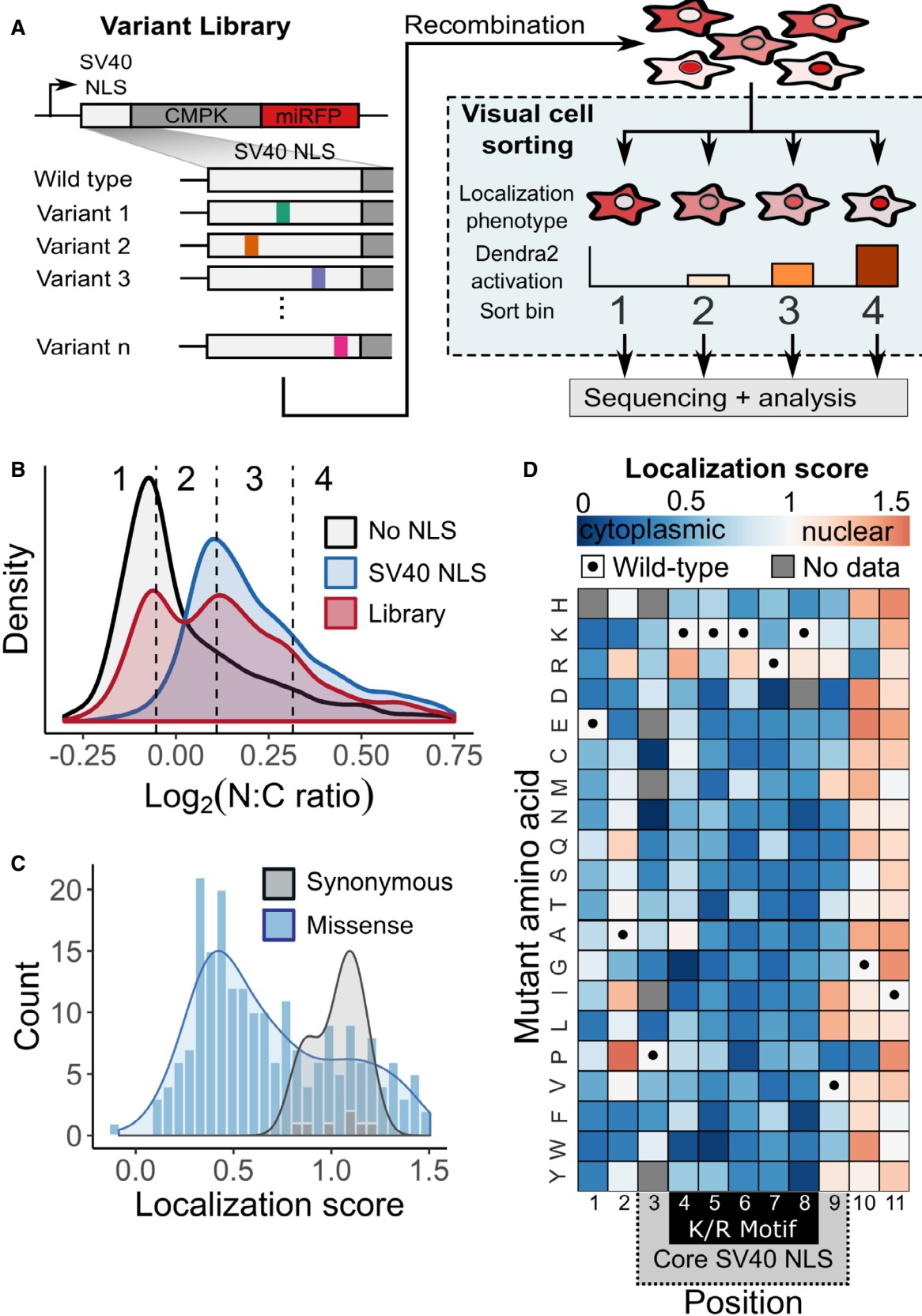

**Figure 2.**

**Figure 2. Visual Cell Sorting for pooled, image-based genetic screening.**

A  A mutagenized simian virus (SV) 40 NLS library containing 346 unique nucleotide variants fused to a chicken muscle pyruvate kinase (CMPK) miRFP reporter was recombined into a U-2 OS H3-Dendra2 cell line. Visual Cell Sorting was performed to separate the NLS library expressing cells into four photoactivation bins according to the microscope-derived nucleus-to-cytoplasm ratio of the miRFP reporter. Each bin was deeply sequenced and analyzed to assign each amino acid variant a quantitative nuclear localization score.

B  U-2 OS H3-Dendra2 cells expressing either the NLS library, a wild-type control, or a no NLS control were imaged at 20× magnification and nucleus-to-cytoplasm (N:C) ratios measured. Curves, estimated kernel density of cells (n = 1,529, 3,269, and 3,931 cells for no NLS, SV40 NLS, and WT NLS, respectively); dotted lines, Visual Cell Sorting photoactivation gates with associated bin numbers.

C  Raw variant nuclear localization scores were calculated using a scaled weighted average of variant frequencies across the four sort bins. WT-like variants have a score of 1 and cytoplasm-localized variants a score of 0. Localization score, mean values of normalized scores from 5 replicates (n = 637,605 cells); curves, kernel density estimate of variant score distributions.

D  Nuclear localization scores of missense variants (n = 202) displayed as a heatmap. Gray boxes, variants not observed or scored in a single replicate; black dots, WT sequence; dotted gray area on the horizontal axis, SV40 NLS often used to localize recombinant proteins to the nucleus; black box, the five residue K/R-rich region.

localization was calculated using a nucleus-to-cytoplasm miRFP intensity ratio (N:C ratio; Fig EV2A). In contrast to the wild-type SV40 NLS-tagged reporter, cells expressing an untagged reporter had a low nucleus-to-cytoplasm ratio (Fig 2B).

We generated a library of 346 NLS nucleotide variants, corresponding to all possible 209 single amino acid missense variants. Cells expressing the library had a bimodal nucleus-to-cytoplasm ratio distribution, indicating that some variants preserved reporter nuclear localization while others disrupted its localization to different degrees (Fig 2B). We divided the library into four photoactivation levels spanning the nucleus-to-cytoplasm ratio range and used Visual Cell Sorting to sort cells into four bins (Fig 2B, dotted lines). A total of 637,605 cells were sorted across 5 replicates (Table EV2). Microscopy on the sorted cells revealed that Visual Cell Sorting faithfully separated cells by the nuclear localization phenotype (Fig EV2B and C). Deep sequencing revealed the frequency of each variant in every bin, and we used these frequencies to compute a quantitative nuclear localization score for 97% of the 209 possible single missense variants (Dataset EV1) (Rubin *et al*, 2017). Scores were subsequently normalized such that wild type had a normalized score of 1 and the bottom 10% of scoring variants had a median normalized score of 0.

As expected, nuclear localization scores for synonymous variants were close to a wild type-like score of one, and most missense scores were lower than one, indicating loss of nuclear localization sequence function (Fig 2C). Furthermore, the SV40 NLS was most sensitive to substitutions in its K/R motif (Fig 2D). Localization scores were reproducible (mean $r = 0.73$; Fig EV2D), and individually assessed nucleus-to-cytoplasm ratios were highly correlated to the localization scores derived using Visual Cell Sorting ($r^2 = 0.91$; Fig 3A). Finally, localization scores of individual variants were correlated with previously reported *in vitro* $K_d$ values for binding to importin alpha ($r = -0.76$, Fig EV3A). Thus, Visual Cell Sorting accurately quantified the effect of NLS variants on their nuclear localization function.

The SV40 NLS is commonly used to localize recombinant proteins to the nucleus and is included in over 10% of all constructs deposited in Addgene (accessed June 2019). Thus, an optimized NLS could improve a wide range of experiments including CRISPR-mediated genome editing. We further investigated three variants that appreciably increased nuclear localization of the reporter compared to the wild-type SV40 NLS. Individually, these variants modestly improved nuclear localization, and a "superNLS" with three missense variants increased nuclear localization by 2.3-fold (Fig 3B and C).

Most NLS prediction algorithms use naturally occurring, individually validated NLS sequences to identify similar sequences in new proteins. By contrast, our data comprise a comprehensive set of NLS-like sequences with variable function. We trained a linear regression model to predict whether any given 11-mer functions as a monopartite NLS by using the experimentally determined amino acid preferences (Bloom, 2014) at each NLS position, which were calculated with the localization score data. We evaluated our model using a test dataset, not used for training, of 30 NLSs in 20 proteins. The resulting model more accurately predicted NLSs than two previously published linear motif scoring models (Nguyen Ba *et al*, 2009; Lin & Hu, 2013), particularly at a stringency where the majority of NLSs are detected (Fig 3D). We used our model to annotate NLSs in nuclear human proteins (Thul *et al*, 2017) according to two score thresholds: one for high-confidence monopartite NLS (precision 0.88, recall 0.23) and one for candidate monopartite NLSs (precision 0.51, recall 0.76). In total, we annotated 2,352 high-confidence monopartite NLSs and an additional 19,909 candidate monopartite NLSs across 6,718 human nuclear proteins (Dataset EV2).

To substantiate that these represent bona fide NLS sequences, we compared the top-scoring 11-mers in exclusively nuclear proteins to those in exclusively cytoplasmic proteins (Fig EV3B and C). As expected, nuclear proteins had higher top-scoring 11mer sequences than cytoplasmic proteins (Wilcoxon rank sum *P*-value $< 10^{-16}$). Twenty-eight percent of the nucleus-only proteins contained an 11-mer with an NLS score higher than our high-confidence cutoff; only 11% of cytoplasmic proteins contained such a sequence. These results are consistent with our predictor identifying monopartite, SV40-like NLSs in the human proteome.

## Visual cell sorting enables transcriptome profiling on image-based phenotypes

To test whether Visual Cell Sorting enables transcriptomics on cells with distinct image-based phenotypes, we performed single-cell RNA sequencing on cells undergoing divergent morphologic responses to paclitaxel. Paclitaxel is a chemotherapeutic agent that stabilizes microtubules and has been used to treat cancer for decades (Rowinsky *et al*, 1992). Even in a clonal population, a subset of cells adopt a lobulated nuclear morphology when treated with a low dose ($\leq 10$ nM) of paclitaxel (Theodoropoulos *et al*, 1999). We treated a telomerase-immortalized cell line derived from human retinal pigment epithelium, hTERT RPE-1, with paclitaxel and observed mitoses that sometimes resulted in nuclear lobulation

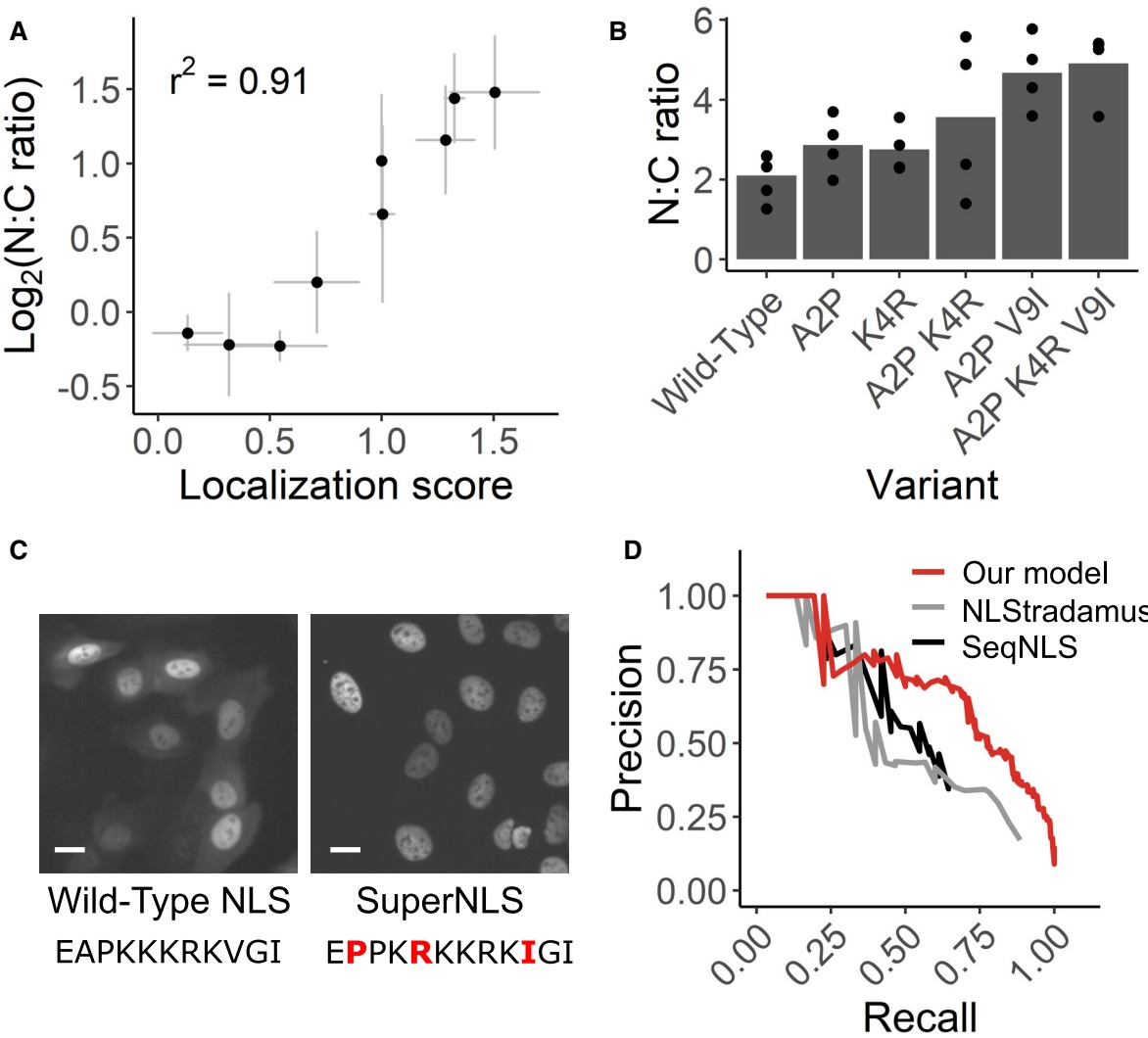

**Figure 3.  Visual Cell Sorting-derived variant scores accurately predict NLS function.**

A   Nine NLS variants were individually expressed in the CMPK-miRFP reporter in U-2 OS H3- Dendra2 cells. The median nucleus-to-cytoplasm (N:C) ratio of cells expressing each variant was measured by microscope and compared to its localization score derived by Visual Cell Sorting. $n \geq 141$ cells per variant per replicate. Bars, mean across at least three separate replicates.

B   SV40 NLS variants that appeared to enhance nuclear localization were individually tested both alone and in combination. NLS variants with up to three amino acid changes were expressed in U-2 OS H3-Dendra2 cells and imaged; the median N:C ratio was quantified across cells in the same well. $n \geq 527$ cells per variant per replicate.

C   Representative images from cells expressing the wild-type SV40 NLS or the optimized superNLS fused to the miRFP reporter. Scale bars = 20 μm; red letters, amino acid differences from the wild type construct.

D   Nuclear localization scores derived from Visual Cell Sorting were used to generate a predictive model that was trained on UniProt NLS annotations. Precision/recall curves for our model and two other linear motif scoring models, NLStradamus (Nguyen Ba *et al*, 2009) and SeqNLS (Lin & Hu, 2013), on a test dataset ($n = 30$ NLSs) are shown.

that persists through the cell cycle (Movies EV1 and EV2). In order to computationally define a cutoff for lobulated nuclei, we measured the shape factor, a circularity metric (Fig 4A), of nuclei in vehicle-treated cells and found that 95% of these morphologically normal cells have a nuclear shape factor > 0.65. We then analyzed pacli-taxel-treated cells and found that 30% of paclitaxel-treated cells had lobulated nuclei, defined by shape factor of < 0.65 (Fig 4B).

Given that morphologic phenotypes are potent indicators of cell state (Rohban *et al*, 2017), we hypothesized that the change in nuclear morphology was accompanied by a distinct gene expression program. To test this hypothesis, we used Visual Cell Sorting to separate morphologically normal paclitaxel-treated cells (shape factor > 0.65) from those with lobulated nuclei (shape factor < 0.65). We then subjected each population of cells to single-cell RNA sequencing (Fig 4C). Imaging, analysis, photoactivation, and FACS-based recovery (Fig EV4A) of ~200,000 cells took < 7 h. Following FACS, we prepared sequencing libraries for approxi-mately 6,000 single-cell transcriptomes from each population. We observed an RNA sequencing batch effect that was completely attri-butable to different levels of cell-free RNA (Cao *et al*, 2017; preprint:

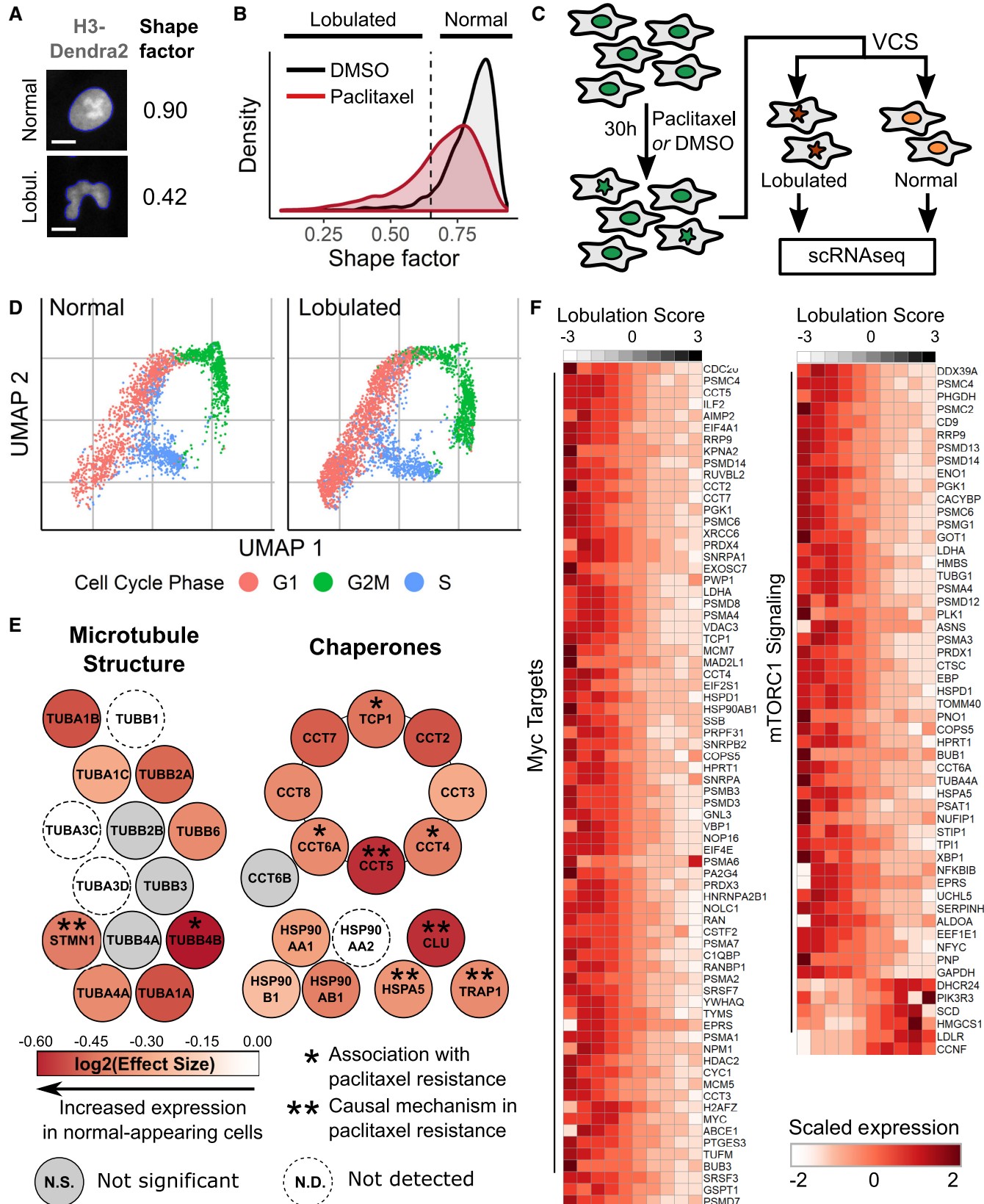

Figure 4.

◀

**Figure 4. Visual Cell Sorting to dissect heterogeneous nuclear morphology following paclitaxel treatment.**

A  RPE-1 NLS-Dendra2 × 3 cells were treated for 24 h with 0.25 nM paclitaxel or DMSO and imaged. The shape factor, which measures the degree of an object's circularity, was computed for each nucleus. One normal nucleus with a shape factor near one and one lobulated nucleus with a low shape factor are shown. The computationally determined boundaries of each nucleus are shown in blue; scale bar = 10 μm.

B  Shape factor density plots for vehicle (DMSO) and 0.25 nM paclitaxel-treated RPE-1 cells ($n \geq 3{,}914$ cells per treatment). Dashed line, cutoff for lobulated nuclei (shape factor < 0.65).

C  RPE-1 cells were treated with 0.25 nM paclitaxel, then subjected to Visual Cell Sorting according to nuclear shape factor. Populations of cells with normal or lobulated nuclei were subjected separately to single-cell RNA sequencing.

D  UMAP analysis of single-cell RNA sequencing results of paclitaxel-treated cells. Expression of cell cycle-related genes was used to annotate each cell as being in G1, S, or G2/M.

E  A differential gene test was performed using as covariates cell cycle scores and a lobulation score, which is higher in lobulated cells compared to morphologically normal cells (Fig EV4D). Genes related to microtubule structure or various chaperone complexes are colored according to the expected $\log_2$ fold change per unit increase in lobulation score (effect size); asterisks, genes associated with paclitaxel resistance (Alli *et al*, 2007; Ooe *et al*, 2007; Di Michele *et al*, 2009; Su *et al*, 2009; Li *et al*, 2013; Dorman *et al*, 2016).

F  Expression counts for genes associated with c-Myc and mTORC1 signaling were aggregated across cells binned according to their lobulation score, then log-normalized and rescaled. Higher lobulation scores correspond to a higher likelihood of nuclear lobulation.

Young & Behjati, 2018) in the lobulated and morphologically normal cell sequencing preps (Fig EV4B).

We used UMAP (preprint: McInnes *et al*, 2018) to visualize a low-dimensional embedding of the single-cell transcriptomes. The distributions of normal and lobulated cells in the UMAP embedding were similar, indicating modest differences in their transcriptomic states. Differences in cell cycle phase (Butler *et al*, 2018) largely explained transcriptomic variation (Fig 4D). More lobulated cells than normal cells were in G1 (53% vs. 44%), suggesting that lobulated cells had an increased propensity to arrest in G1. Indeed, G1 arrest is known to occur after paclitaxel treatment in non-transformed cell lines (Trielli *et al*, 1996).

To understand the relationship between transcriptomic variation, lobulation, and cell cycle, we examined the top batch-corrected principal components of the single-cell transcriptomes. We noticed that the first four principal components separated cells by nuclear morphology (Fig EV4C). To discover the genes associated with nuclear lobulation while controlling for the cell-free RNA batch effect, we sequenced an unseparated paclitaxel-treated cell population and aligned their transcriptomes to those from morphologically normal and lobulated cells (Haghverdi *et al*, 2018). We then derived a lobulation score for each cell via linear combinations of the four principal components that correlate with nuclear morphology (Fig EV4D). Finally, we extracted genes associated with this lobulation score, which is higher in cells with lobulated nuclei, in the unseparated cells by using a differentially expressed gene test (Materials and Methods).

In total, 765 genes were significantly associated with the lobulation score (adjusted *P*-value < 0.01; Fig EV4E, Dataset EV3). To our surprise, the vast majority (84%) of these genes were more highly expressed by morphologically normal cells. Morphologically normal cells upregulated the genes encoding actin and microtubules (e.g. *ACTB*, *TUBB4B*; Figs 4E and EV4F), a well-documented response to microtubule damage and paclitaxel treatment (Gasic *et al*, 2019). We also noted that these cells upregulated the chaperone clusterin (*CLU*) and its co-activator *HSPA5*, which together decrease paclitaxel-mediated apoptosis by stabilizing mitochondrial membrane potential (Li *et al*, 2013). Intrigued by the notion that morphologically normal cells are resisting the effects of paclitaxel, we searched the literature for other genes upregulated in these cells and found that many of them, including *PRMT1* (Cho *et al*, 2012), *ENO1* (Georges *et al*, 2011), *STMN1* (Alli *et al*, 2007), *LDHA* (Zhou *et al*,

2010), *ANXA5* (Di Michele *et al*, 2009), and *HSPA8* (Sugimura *et al*, 2004), are associated with paclitaxel resistance in diverse cancers.

To better understand the gene expression program associated with normal nuclear morphology in the context of paclitaxel treatment, we looked for enrichment of genes in previously defined gene sets (Liberzon *et al*, 2015) covering a host of cellular processes (Dataset EV4 and EV5). Morphologically normal cells upregulated 7 out of 8 proteins in the chaperonin containing TCP-1 complex (adjusted *P*-value = 7.64e-15; Fig 4E), which is critical for tubulin folding and has been previously associated with paclitaxel resistance in ovarian cancer (Di Michele *et al*, 2009). Morphologically normal cells also upregulated the transcriptional targets of two paclitaxel resistance-associated signaling pathways (Shafer *et al*, 2010; Parasido *et al*, 2019): c-Myc (adjusted *P*-value = 1.66e-30) and mTORC1 (adjusted *P*-value = 6.19e-17; Fig 4F). Together, these results suggest that the morphologically normal, paclitaxel-treated cells exhibit a biosynthetic and proteostatic gene expression program, with remarkable similarities to the gene expression profiles observed in paclitaxel-resistant cell lines and cancers.

## Discussion

A major limitation of current microscopy-based experiments is the inability to isolate hundreds of thousands of phenotypically defined cells for further analysis. We developed Visual Cell Sorting, a microscope-based method that directs a digital micromirror device to irreversibly photoactivate a genetically encoded fluorescent protein in cells of interest, effectively translating a complex visual phenotype into one that can be sorted by FACS.

To highlight the Visual Cell Sorting's flexibility, we performed two distinct experiments. First, we leveraged its high throughput to quantify the function of hundreds of nuclear localization sequence variants in a pooled, image-based genetic screen. By combining single variants that individually improved NLS function, we created an eleven-residue superNLS (EPPRKKRKIGI) that could be used to improve CRISPR-mediated genome editing, fluorescent protein-based nuclear labeling, and other experiments that leverage nuclear recombinant proteins. We then used the variant scores to make an accurate, amino acid preference-based predictor of NLS function, which we applied to the human nuclear proteome and validated by comparing the top-scoring sequences between cytoplasmic and

nuclear proteins. Interestingly, some cytoplasmic proteins contain putative NLSs, which could be explained by an NLS that becomes accessible to the nuclear import machinery after a signaling event (Beg *et al*, 1992) or a nuclear export signal located on the same protein that overwhelms an otherwise functional NLS (Marchand *et al*, 2019). Nuclear proteins without high-scoring sequences may harbor a non-SV40 type NLS or have an interaction partner with a functional NLS enables co-import into the nucleus.

In a second application, we leveraged Visual Cell Sorting's ability to recover live, phenotypically defined subsets of cells to investigate the heterogeneous cellular response to paclitaxel treatment using single-cell RNA sequencing. To our surprise, cells that resist the effect of paclitaxel on nuclear morphology appear to be counteracting the drug's effects at the molecular level with a gene expression program similar to paclitaxel-resistant cancers. This phenomenon, whereby a subset of clonal cells resists the effects of drug treatment with a protective gene expression program, is reminiscent of the "pre-resistance" reported in primary melanoma cells (Shaffer *et al*, 2017). However, the experiment we conducted cannot determine whether this gene expression program pre-exists in the population or is stochastically induced upon paclitaxel addition. To answer this question, live-cell microscopy or cell barcoding could be used to determine whether pre-treatment levels of the genes expressed highly in morphologically normal cells (e.g. *TUBB4B* expression, c-Myc targets) leads to morphologic responses and survival after paclitaxel treatment.

High throughput is a key advantage of Visual Cell Sorting, compared to other similar methods. In our pooled image-based screen, we analyzed approximately one million cultured human cells across 60 h of imaging and sorting time, ultimately recovering ~650,000. This throughput is ~1,000-fold more than what could be achieved using other photoconvertible fluorophore-based methods (Chien *et al*, 2015; Binan *et al*, 2016, 2019; Kuo *et al*, 2016), ~20-fold more than current MERFISH pooled screens (Wang *et al*, 2019), and similar in per-day throughput to *in situ* sequencing-based screens (Feldman *et al*, 2019; Table EV1). Thus, Visual Cell Sorting enables the analysis of thousands of genetic variants in a single experiment. Visual Cell Sorting throughput could be increased even further by analyzing cellular phenotypes at a lower magnification, by applying faster image analysis algorithms, or by shutting off Dendra2 expression before imaging to extend imaging time (Fig EV1B).

A second key advantage of Visual Cell Sorting is that it does not require any expensive dye-based reagents such as oligo libraries or fluorescent-labeled oligos, customized hardware components, or complex workflows. Outfitting an automated wide-field microscope requires just three inexpensive, commercially available components: a live-cell incubation chamber, a digital micromirror device, and a 405 nm laser. Finally, Visual Cell Sorting enables recovery of cells with up to four distinct phenotypes in one experiment, unlike other photoconvertible fluorophore-based methods (Chien *et al*, 2015; Binan *et al*, 2016, 2019; Kuo *et al*, 2016; David *et al*, 2017).

Visual Cell Sorting has important limitations. Cells must be genetically engineered to express Dendra2, which is photoactivated by blue fluorescent protein (BFP) excitation wavelengths and emits at GFP and RFP wavelengths. This requirement limits the other fluorescent channels are available for imaging. However, miRFP (Shcherbakova *et al*, 2016) and mBeRFP (Yang *et al*, 2013) can be used in conjunction with Dendra2, allowing two additional compartments or proteins to be marked in each experiment. Moreover, new analytical approaches leveraging brightfield images may reduce the need for fluorescent markers (Christiansen *et al*, 2018; Ounkomol *et al*, 2018). Another limitation is that, unlike morphological profiling approaches (Bray *et al*, 2016), Visual Cell Sorting requires a predefined phenotype of interest and is limited by current FACS hardware to 4 phenotypic bins. Finally, Visual Cell Sorting experiments are limited to approximately twelve hours to avoid Dendra2 activation signal decay or cell overgrowth. The several hours required to execute a Visual Cell Sorting experiment makes it challenging to study transient phenotypes (e.g. cell cycle-dependent phenotypes). Furthermore, decay of photoactivated Dendra2 may be more pronounced in rapidly dividing bacterial or yeast as activated Dendra2 is diluted by cell division. However, the workflow we present, with imaging at 20× magnification and image processing times of 3–8 s, is sufficient for the analysis of hundreds of thousands of human cells in one experiment.

In summary, Visual Cell Sorting is a robust and flexible method that can be used to separate heterogeneous cultures of cells into up to four morphologically defined subpopulations. The components required for Visual Cell Sorting are already in widespread use, are commercially available, and can be adapted to most modern automated wide-field fluorescent microscopes. The method will improve in scope and speed as further advances are made in cell segmentation and image analysis. We demonstrate that Visual Cell Sorting can be used for both image-based pooled genetic screens and image-based transcriptomics experiments. This flexibility should drive the application of Visual Cell Sorting to a wide range of biological problems in diverse fields of research that seek to dissect cellular heterogeneity, including stem cell biology, functional genomics, and cellular pharmacology.

# Materials and Methods

**Reagents and Tools table**

| Reagent/Resource | Source | Identifier or Catalog number |
|---|---|---|
| **Experimental Models** | | |
| U-2 OS cells | ATCC | HTB-96 |
| hTERT RPE-1 cells | ATCC | CRL-4000 |
| HEK 293T cells | ATCC | CRL-3216 |

**Reagents and Tools table** (continued)

| Reagent/Resource | Source | Identifier or Catalog number |
|---|---|---|
| **Recombinant DNA** | | |
| Dendra2-Lifeact7 | Addgene | 54694 |
| mEmerald-H3-23 | Addgene | 54115 |
| pH2B-miRFP703 | Addgene | 80001 |
| psPAX2 | Addgene | 12260 |
| pMD.2 | Addgene | 12259 |
| pLenti CMV rtTA3 Blast | Addgene | 26429 |
| Other constructs, gBlocks, etc. | This study | Table EV3 |
| **Antibodies** | | |
| None | NA | NA |
| **Oligonucleotides and sequence-based reagents** | | |
| PCR primers | This study | Table EV3 |
| **Chemicals, enzymes and other reagents** | | |
| KAPA Hifi 2× polymerase | Kapa Biosystems | KK2601 |
| Dulbecco's modified Eagle's medium (DMEM) | Thermo Fisher | 11965118 |
| DMEM, no phenol red | Thermo Fisher | 21063045 |
| DMEM/F12 | Thermo Fisher | 11320033 |
| DMEM/F12, no phenol red | Thermo Fisher | 21041025 |
| Doxycycline | Sigma | D9891 |
| Trypsin−EDTA 0.25% | Thermo Fisher | 25200056 |
| OPTIMEM | Fisher Scientific | 31985070 |
| FuGENE6 | Promega | E2691 |
| 2X Gibson Assembly Master Mix | NEB | E2611L |
| DpnI | NEB | R0176L |
| DNA Clean & Concentrator | Zymo Research | D4013 |
| GenElute HP Plasmid DNA Midiprep Kit | Sigma | NA0200-1KT |
| PEG-it Virus Precipitation Solution | SBI | LV810A-1 |
| Lipofectamine 3000 | Thermo Fisher | L3000015 |
| **Software** | | |
| Metamorph (v7.10.1.161) | Molecular Devices | |
| **Other** | | |
| Leica DMi8 with Adaptive Focus | Leica | |
| Incubation i8 chamber | Leica | |
| TempController 2000-1 | PeCon | |
| CO2 regulator | Oko | |
| Spectra X Light Engine LED | Lumencor | |
| Multi-band dichroic filter | Spectra Services | LED-DA-FI-TR-Cy5-4X-A-000 |
| Multi-band dichroic filter | Spectra Services | LED-CFP/YFP/mCherry-3X-A-000 |
| Bright-line band-pass filter (DAPI) | Semrock | FF01-433/24-25 |
| Bright-line band-pass filter (GFP) | Semrock | FF01-520/35-25 |
| Bright-line band-pass filter (RFP) | Semrock | FF01-600/37-25 |
| Bright-line band-pass filter (NIR) | Semrock | FF01-680/42-25 |
| 20 × 0.8 NA apochromatic objective | Leica | |
| Mosaic 3 Digital Micromirror Device | Andor | |
| Mosaic SS 405/1.1 W laser | Andor | |

Reagents and Tools table (continued)

| Reagent/Resource | Source | Identifier or Catalog number |
|---|---|---|
| iXon Ultra 888 EMCCD monochrome camera | Andor | |
| Glass-bottom black-walled plates | CellVis | P06-1.5H-N |
| LSR-II | BD Biosciences | |
| FACS Aria III | BD Biosciences | |

## Methods and Protocols

### General reagents, DNA oligonucleotides, and plasmids

Unless otherwise noted, all chemicals were obtained from Sigma and all enzymes were obtained from New England Biolabs (Ipswich, MA). KAPA Hifi 2x Polymerase (Kapa Biosystems; Wilmington, USA; cat. no. KK2601) was used for all cloning and library production steps. E. coli were cultured at 37°C in Luria broth. All cell culture reagents were purchased from Thermo Fisher Scientific (Waltham, MA) unless otherwise noted. HEK 293T cells (ATCC; Manassas, VA; CRL-3216) and U-2 OS cells (ATCC HTB-96), and derivatives thereof were cultured in Dulbecco's modified Eagle's medium supplemented with 10% fetal bovine serum, 100 U/ml penicillin, 0.1 mg/ml streptomycin, and 1 μg/ml doxycycline (Sigma; St. Louis, MO), unless otherwise noted. hTERT RPE-1 cells (ATCC CRL-4000) and derivatives thereof were cultured in F12/DMEM supplemented with 10% FBS, 1 mM PenStrep, and 0.01 mg/ml hygromycin B. For Visual Cell Sorting experiments, DMEM without phenol red was used to reduce background fluorescence. Cells were passaged by detachment with Trypsin–EDTA 0.25%. All cell lines tested negative for mycoplasma in monthly tests. All synthetic oligonucleotides were obtained from IDT, and their sequences can be found in Table EV3. All non-library-related plasmid modifications were performed with Gibson assembly. See the Appendix and Table EV3 for construction of the vectors used.

### Construction of the SV40 NLS library

A site-saturation mutagenesis library of the SV40 NLS upstream of a tetramerizing miRFP reporter (attB-NLS-CMPK-miRFP library) was constructed using Gibson cloning (Gibson et al, 2009). See the Appendix for a detailed description of the construction of the site-saturation mutagenesis library.

### Cell lines

U-2 OS cells (ATCC, HTB-96) expressing the Tet-ON Bxb1 landing pad (U-2 OS AAVS-LP Clone 11) were generated as previously described (Matreyek et al, 2017). To create H3-Dendra2- and H3-Dendra2/H2B-miRFP-expressing derivative cell lines, attB-H3-Dendra2 or attB-H3-Dendra2-P2A-H2B-miRFP703 were recombined into U-2 OS AAVS-LP Clone 11 cells, as previously described (Matreyek et al, 2017). For the NLS work, a separate clonal U-2 OS cell line expressing the Tet-ON landing pad and CMV-H3-Dendra2 was created by co-transduction of parental U-2 OS cells with the LLP-Blast lentivirus (Matreyek et al, 2020) and another expressing histone H3-Dendra2 (U-2 OS LLP-Blast/H3-Dendra2 Clone 4). A clonal hTERT RPE-1 cell line expressing CMV-NLS$_{SV40}$-Dendra2-GSSG-Dendra2-GSSG-Dendra2 (NLS-Dendra2x3); CMV-H2B-miRFP; and CMV-NES-mBeRFP was generated by transduction of a parental line (ATCC, CRL-4000) with three lentiviral vectors followed by single-cell sorting (RPE-1 NLS-Dendra2x3/H2B-miRFP/NES-mBeRFP Clone 3). For more information regarding these lines and for the lentiviral production protocol, see the Appendix.

### Recombination of single-variant SV40 NLS clones or the library into U-2 OS LLP-Blast/H3-Dendra2 Clone 4 cells

The SV40 NLS variant library or single-variant clones were recombined into U-2 OS LLP-Blast/H3-Dendra2 Clone 4 cells, as previously described in HEK 293Ts (Matreyek et al, 2017). Two recombination replicates were performed. For more information, see the Appendix.

### Visual cell sorting: equipment and settings

A Leica DMi8 inverted microscope was outfitted with Adaptive Focus; an Incubator i8 chamber with PeCon TempController 2000-1 and Oko CO$_2$ regulator set to 5%; a 6-line Lumencor Spectra X Light Engine LED; Semrock multi-band dichroic filters (Spectra Services, Ontario, NY; cat. no. LED-DA-FI-TR-Cy5-4X-A-000, LED-CFP/YFP/mCherry-3X-A-000); bright-line band-pass emissions filters for DAPI (433/24 nm), GFP (520/35 nm), RFP (600/37 nm), and NIR (680/22 nm); a 20 × 0.8 NA apochromatic objective; and a Mosaic3 Digital Micromirror Device affixed to a Mosaic SS 405 nm/1.1 W laser and mapped to an iXon 888 Ultra EMCCD monochrome camera. The microscope and digital micromirror device were controlled with the Metamorph Advanced Image Acquisition software package (v7.10.1.161; Molecular Devices, San Jose, CA). The image size was ~560 × 495 μm. Image bit depth ranged from 12 to 16 bits, depending on the brightness of cells in the field of view.

Cells were plated and imaged on glass-bottom, black-walled plates (CellVis, Mountain View, CA; P06-1.5H-N, P24-1.5H-N, P96-1.5H-N) in phenol red-free media at 5% CO$_2$ and 37°C using the 20 × 0.8 NA objective. ~50–100 cells were imaged per field of view. To image unactivated Dendra2, 474/24 nm excitation and 482/25 nm emission filters were used. To image activated Dendra2, 554/23 nm excitation and 600/37 nm emission filters were used. To image miRFP, 635/18 nm excitation and 680/22 nm emission filters were used. Prior to imaging, the Auto Focus Control system was activated. Metamorph's Plate Acquisition module was used to collect images and run Metamorph journals that analyzed cells and directed their selective photoactivation by the digital micromirror device. For more information about the Metamorph journals used to image and activate cells, see the Appendix.

### Visual Cell Sorting: cell preparation, imaging, analysis, and photoactivation

An up-to-date version of this protocol can be found at protocols.io (https://www.protocols.io/view/visual-cell-sorting-beigjcbw).

1  24–48 h before imaging, plate cells onto 6-well glass-bottom, black-walled plates at a density of 50,000–200,000 cells per well.

2  Before imaging, wash cells with 1× DPBS and add complete media without phenol red.

3  Turn on the microscope and incubation chamber, set the $CO_2$ regulator to 5%, and open Metamorph.

4  Place cells in microscope and bring cells into focus. Test imaging conditions (LED power, exposure time, etc.) for the desired channels.

5  Turn on Auto Focus Control. Using the Well Plate Acquire dialog box, image ~25–100 sites of experimental conditions (and controls, if applicable). Initialize a log file to collect phenotypic data. Using the Journal > Loop > Loop Through Images in Directory command, run the analysis journal on the images to collect the desired phenotypic information. The journal must include an "Integrated Morphometry—Measure" or a "Region Measurements" command to add phenotypic information for each cell to the log file. *Note:* These specific images will not be used for activation; rather, this analysis serves to ensure that the phenotypes match what one would expect.

6  Save the imaging conditions used for the Well Plate Acquire dialog box as a state file.

7  Close the log file. Check the distribution of phenotypes in experimental conditions and controls by running custom software (e.g. Python script) with the log file as input.

8  Load the site map. As of Metamorph v7.10.1.161, this must be done by:
   a. Closing Metamorph
   b. Replacing the *htacquir.cfg* file in the Metamorph application Groups > Metamorph directory with an *htacquir.cfg* file that contains the site map. *htacquir.cfg* files that contain various site maps for 6- and 24-well plates used in our experiments can be found on the GitHub repository under the Metamorph directory.
   c. Reopening Metamorph and reloading the saved state file (load everything except for site map settings). *Note:* In Metamorph v7.10.1.161, the site map can be contaminated by extra sites in the top left corner after this operation. Check the "Sites" tab of the Well Plate Acquire dialog box and remove any extra sites by left clicking.

9  Center the well:
   a. Move the objective to the approximate center of well A1.
   b. Under the Well Plate Acquire "Plate" tab, select "Set A1 Center . . ." > "Set A1 Center to Current".
   c. Under the "Sites" tab, move the objective to the top center site by right clicking.
   d. Using the eyepiece and brightfield illumination settings, check whether the objective is centered at the top of the well. If not, manually change the A1 center settings (measured in microns) to move it in the desired direction.
   e. Repeat steps (D) and (E) until the top center site of the site map is centered on the top.
   f. Re-check that cells are in focus and that Auto Focus Control in "on". Auto Focus Control can be turned off by the objective moving too far from the plate and hitting the plate holder.

10  Select the wells to be subject to Visual Cell Sorting under the "Plates" tab by left clicking

11  Select appropriate journals to be run at the Start of Plate, After Imaging, and End of Plate under the "Journals" tab
   a. The "Start of Plate" journals (labeled "startup.jnl" in the GitHub repository) serve to add a delay to imaging, if necessary; set the 405 nm pulse times for the activations; and set any phenotypic threshold values (e.g. NC ratios) for activation.
   b. The "After Imaging" journals contain analysis and activation scripts that are performed after each image is taken
   c. The "End of Plate" journals turn off the laser to increase its lifetime

12  OPTIONAL: Re-align the digital micromirror device:
   a. Under Devices > Mosaic Targeted Illumination, click "Update Settings" in the Configuration tab
   b. Follow the instructions to re-calibrate the device

13  OPTIONAL: Run the experiment without the laser on to check that the correct cells are being identified and activated:
   a. In the Well Plate Acquire dialog box, hit "Acquire"
   b. Watch the first 5–10 sites of imaging, analysis, and marking cells for activation. In the activation journals associated with this publication, nuclei subject to the three activation states (50, 200, and 800 ms) are outlined in three different colors.

14  Turn on the laser

15  Hit "Acquire" to begin acquisition, analysis, and activation.

### Visual cell sorting: FACS on microscope-activated cells

Cells activated on the microscope were analyzed using an LSR-II (BD Biosciences; San Jose, CA) or sorted into bins according to their Dendra2 photoactivation state using a FACS Aria III (BD Biosciences). Raw.fcs files and code associated with this work are available on GitHub. For more information, see the Appendix.

1  Trypsinize cells and resuspend in DPBS supplemented with 1–2% FBS or BSA

2  Make a gate for live cells using a SSC-A vs. FSC-A plot.

3  Within the live-cell gate, make a gate for single cells using a FSC-W vs. FSC-A plot.

4  Within the single-cell gate, make a gate for Dendra2-positive cells using a FITC-A histogram plot. In some clonally derived lines, Dendra2 expression will silence over the course of weeks to months. If Dendra2-negative cells exceed 10%, we recommend resorting the population or returning to a lower passage stock.

5  Create an activated (PE-YG-A) vs. unactivated (FITC-A) Dendra2 scatter plot. Draw gates for the activated populations of interest. Activated populations will appear as diagonal clouds with higher PE-YG-A signals than a negative control.

6  Create a ratio (PE-YG-A/FITC-A) histogram. Show the activated populations of interest (defined in Step 5) within the ratio histogram. Create sorting gates for each population.

7  Sort populations of activated cells according to the gates set on the ratio histogram plot.

8  Spin cells for 5 min at 300–500 × $g$, then plate cells in warm, complete media.

9  Analyze data using FlowCytometryTools (v0.5.0) in Python (v3.6.5) or flowCore (v1.11.20) in R (v3.6.0).

### Selective photoactivation of cells expressing miRFP

U-2 OS AAVS-LP Clone 11 cells with attB-H3-Dendra2 or attB-H3-Dendra2-P2A-H2B-miRFP recombined into the landing pad were counted and mixed in ratios ranging from 0.5% to 50% miRFP-expressing cells, and then, 40,000 cells of each mixture were seeded into three wells of a 24-well plate. The next day, cells were placed on the microscope and imaged, analyzed, and activated at 661 sites across each well of the plate, covering ~95% of the total well area. At each site, Dendra2 and miRFP were imaged with 2 × 2 binning; Metamorph's Count Nuclei module was used on the miRFP image to identify miRFP-expressing cells; and a binary with regions corresponding to miRFP-expressing cells was passed to the digital micromirror device, which subsequently activated the cells. Once all sites were imaged, analyzed, and activated, the cells were subject to flow cytometry to assess unactivated Dendra2, activated Dendra2, and miRFP expression. The experiment was repeated two additional times for a total of three replicates. For the Metamorph journals used to analyze and activate cells, see the GitHub repository. For more information about the gating scheme used for this experiment, see Appendix Fig S1.

### Photoactivation of cells for 0, 50, 200, and 800 ms

U-2 OS AAVS-LP Clone 11 cells with attB-H3-Dendra2-P2A-H2B-miRFP recombined into the landing pad were seeded at 50,000 cells per well in a 6-well glass-bottomed plate. The next day, cells were imaged for unactivated Dendra2 and miRFP at 100 sites (10 × 10 square) and quartiles of total miRFP intensity were measured using Metamorph. Then, cells across 661 sites in two wells were left unactivated or activated for 50 ms, 200 ms, or 800 ms according to the miRFP intensity quartile to which they belonged (Q1 = 0–3,803, Q2 = 3,804–5,839, Q3 = 7,396–9,674, Q4 = 9,674+). For the Metamorph journals used to analyze and activate cells, see the GitHub repository.

### Testing for photoactivation-induced toxicity with Annexin V and DAPI

U-2 OS AAVS-LP Clone 11 cells with attB-H3-Dendra2 recombined into the landing pad were seeded at 20,000 cells per well in a 24-well plate. Over the next 2 days, cells across 400 sites (60% well coverage) in three replicate wells were segmented using the Count Nuclei module in Metamorph and activated for 800 ms. Forty-eight hours after the first well was activated, cells were trypsinized, stained with Annexin V (Thermo, cat. no. A23204) and DAPI (Invitrogen, cat. no. D1306), and subjected to flow cytometry to assess unactivated Dendra2, activated Dendra2, Annexin V, and DAPI. Three wells of unactivated cells were heated at 50°C for 10 min as a cell death positive control. The experiment was repeated two additional times for a total of three replicates. Data were analyzed using FlowJo (v10.5.3).

### Testing for photoactivation-induced toxicity with RNA sequencing

U-2 OS AAVS-LP Clone 11 cells with attB-H3-Dendra2 recombined into the landing pad were seeded at 20,000 cells per well in 8 wells of a 24-well plate. Eighteen hours later, cells across 6 wells (678 sites per well; ~100% well coverage) were activated and then incubated for 0.5, 1.5, 2.5, 3.5, 4.5, or 6 h (1 well each). Two wells were left unactivated. Dendra2 photoactivation was verified by flow cytometry, and the two unactivated samples were used as negative controls. Bulk RNA sequencing libraries were prepared as described previously (Cao et al, 2017). Briefly, RNA was extracted from each sample using a Trizol/RNeasy Mini Kit (Thermo Fisher, cat. no. 15596026, Qiagen; Germantown, MD; cat. no. 74104) then subjected to SuperScript IV First-Strand Synthesis (Thermo Fisher 18091050) and NEBNext Ultra II Directional RNA Second Strand Synthesis (NEB E7550), according to the manufacturer's instructions. cDNA was then tagmented with Nextera Tn5 (Illumina; San Diego, CA; FC-131-1024) and amplified/indexed by PCR with the NEBNext DNA Library Prep Kit (NEB E6040). Samples were sequenced using a NextSeq 500/550 75 cycle kit (Illumina, cat. no. TG-160-2005). Differential gene expression analysis of RNA sequencing data followed the standard DESeq2 workflow (Love et al, 2014). Briefly, differential gene expression testing was performed using a binary coding of photoactivation status in the DESeq2 design formula. Dispersion estimates, $\log_2$ fold changes and adjusted P-values were all calculated using the DESeq () function with default parameters as specified in DESeq2.

### Visual cell sorting of cells expressing SV40 NLS library

Eighteen hours before imaging, 300,000 U-2 OS LLP-Blast/H3-Dendra2 Clone 4 cells with the attB-NLS-CMPK-miRFP library recombined into the landing pad were seeded into each well of a 6-well plate. The next day, cells were placed onto the microscope and imaged, analyzed, and activated across 2,949 sites (~100% well coverage) across two wells. At each site, Dendra2 and miRFP were imaged with 2 × 2 binning; Metamorph's Count Nuclei module was used on the Dendra2 image to identify nuclei and create a nuclear binary image; cytoplasm binaries were created by subjecting the nuclear binary to a dilate function and subtracting away the nuclear binary; each nucleus–cytoplasm binary pair was superimposed on the miRFP image and average pixel intensities were measured for each compartment; cells with an average nuclear or cytoplasmic miRFP pixel intensity of less than 11,000 were filtered out; a nucleus-to-cytoplasm (N:C) ratio was calculated by dividing the average nuclear pixel intensity by the average cytoplasmic pixel intensity; nuclei with N:C < 0.964 were not activated at all, N:C 0.964–1.079 were activated for 50 ms, N:C 1.079–1.244 were activated for 200 ms, and N:C > 1.244 were activated for 800 ms. Once all sites were imaged, analyzed, and activated, the cells were subject to FACS and unactivated Dendra2 (FITC), activated Dendra2 (PE-YG), and miRFP (AlexaFluor-700) fluorescence intensities assessed. Cells were then sorted into four photoactivation bins (Fig 2B). A total of two Visual Cell Sorting technical replicates were performed on recombination replicate 1, and three were performed on recombination replicate 2. The details of replicate sorts for the NLS library can be found in Table EV2. For an example of the gating scheme, see Appendix Fig S2.

### Sorted SV40 NLS library genomic DNA preparation and sequencing

After sorting, cells in each Dendra2 photoactivation bin were grown in the absence of doxycycline until confluent in one well of a 6-well plate (~7 days), then pelleted and stored at −20°C. DNA was extracted from cell pellets with the DNEasy kit (Qiagen, cat. no. 69504) using RNAse according to the manufacturer's instructions. gDNA was amplified using SV40_NLS_seq_f and SV40_NLS_seq_r (Reagents and Tools table) primers using Kapa Hifi (Kapa Biosystems, cat. no. KK2602) according to the manufacturer's instructions.

Amplicons were cleaned using Ampure XP beads (Beckman Coulter; Brea, CA; cat. no. A63880), then subjected to an indexing PCR step using KAPA2G Robust (Kapa Biosystems, cat. no. KK5705) with primers P5 and an indexing primer (Reagents and Tools table). Amplicons were then run on a 1.5% agarose gel at 130 V for 40 min and the DNA in the 235-bp band extracted using Freeze 'N Squeeze DNA Gel Extraction Spin Columns (Bio-Rad, cat. no. 7326165). Extracted DNA was sequenced on an Illumina NextSeq500 using SV40_NLS_Read1, SV40_NLS_Read2, and SV40_NLS_Index1 primers (Reagents and Tools table). Reads were trimmed and merged using PEAR (Zhang *et al*, 2014). Sequences were quality-filtered and variants were called and counted by using Enrich2, as previously described (Rubin *et al*, 2017). The Enrich2 configuration file is available on the GitHub repository.

### Calculating NLS variant localization scores

Jupyter v5.5.0 running Python v3.6.5 was used for analyses of the Enrich2 output. First, two filters were applied to remove low-quality variants: (i) a minimum nucleotide variant count cutoff of 5 in each bin in each replicate and (ii) a requirement that the variant was accessible via NNK codon mutagenesis. After filtering, remaining nucleotide variants encoding the same amino acid substitution were added to yield a sum of counts for that variant within each bin for each replicate. To generate raw quantitative scores ($S_{raw}$), a weighted average approach as previously described (Matreyek *et al*, 2018) was applied to the variant frequencies ($f_{var}$) across the 4 bins (b1–b4) in each replicate:

$$S_{raw} = \frac{0.25(f_{var_{b1}}) + 0.50(f_{var_{b2}}) + 0.75(f_{var_{b3}}) + f_{var_{b4}}}{f_{var_{b1}} + f_{var_{b2}} + f_{var_{b3}} + f_{var_{b4}}}$$

Raw scores were subsequently normalized such that variants with a wild-type raw score ($S_{WT}$) have a normalized score of 1 and variants with the median raw score of the bottom 10% of variants ($S_{P10}$) have a normalized score of 0:

$$S_{norm} = \frac{S_{raw} - median(S_{P10})}{S_{WT}}$$

A final round of frequency filtering for variants, which sought to increase score correlations without excluding too many variants, removed variants present at a frequency lower than 0.003% of reads in all bins. Then, the raw and normalized scores were recalculated for each replicate; and the mean and standard error of the normalized scores from the five replicates were calculated to produce final scores. An iPython notebook file with the code used to run the analysis is available on the GitHub repository.

### Validation of single NLS variants

ssDNA oligos (IDT, Newark, NJ) encoding the NLS variants were introduced into EcoRI-digested attB-EcoRI-CMPK-miRFP reporter plasmid via a Gibson reaction (Gibson *et al*, 2009). Variants were validated by Sanger sequencing. Plasmids were recombined into 80,000 U-2 OS cells in a 24-well plate using 1.5 μl of FuGENE6 (Promega; Madison, WI; cat. no. E2691) in 100 μl OPTIMEM (Fisher Scientific; Waltham, MA; cat. no. 31985070) with 100 ng of pCAG-Bxb1 and 295 ng of the attB variant recombination plasmid. After 5 days, recombined cells, which are miRFP$^+$, were isolated using

FACS for miRFP$^+$ cells and plated in glass-bottom 24-well plates. Then, recombined cells were imaged for H3-Dendra2 and miRFP. Metamorph was used to segment nuclei and calculate mean nuclear and cytoplasmic miRFP intensity for each cell, as described above ("Visual Cell Sorting on cells expressing SV40 NLS library"). miRFP intensities were background-corrected (see Appendix), and cells with nuclear and cytoplasm miRFP intensities roughly equal to background levels were removed. Then, N:C ratios were calculated for each cell using the cell's mean nuclear ($I_{nuc}$) and cytoplasmic ($I_{cyt}$) miRFP intensities:

$$NC = \frac{I_{nuc}}{I_{cyt}}$$

Each variant was examined in at least three separate imaging replicates. For more information regarding the validation of single NLS variants, see the Appendix.

### Prediction of novel human NLS's

Analysis of the normalized variant localization scores was done in RStudio v1.1.456 running R v3.6.0. Position-wise amino acid preferences were calculated (Bloom, 2014):

$$x_{r,a} = \frac{s_{r,a} - min(s_r)}{max(s_r) - min(s_r)}$$

where $x_{r,a}$ is the amino acid preference for amino acid $a$ at position $r$, $s_{r,a}$ is the mean raw score of variants with amino acid $a$ at position $r$, and $s_r$ is the set of all raw scores at position $r$. The scores of missing variants were estimated using the median score at that variant's position. To train a weighted preference model, NLS sequences ($n = 573$) were downloaded from UniProt using a SPARQL query for all human proteins with a sequence motif annotation that contained the string "Nuclear localization" in its comment. A set of 573 "likely NLS" 11mers were generated by repeating the following for each NLS: (i) scoring every 11mer peptide overlapping the annotated NLS sequence by summing the amino acid preferences of the 11mer peptide (ii) annotating the maximum-scoring 11mer as a "likely NLS". All other possible 11mers in the training dataset (333,255 total) were annotated as "no NLS". To account for the fact that some the amino acid preferences at some positions may be more important than others, a linear regression model of the following form was fit to these data:

$$Y = \beta_0 + \sum_{r=1}^{11} \beta_r x_{r,a}$$

where $Y$ denotes the sequence class ("no NLS" = 0, "likely NLS" = 1), $\beta_0$ is the intercept, $\beta_r$ is the weight given to the amino acid preferences at position $r$, and $x_{r,a}$ is the is the preference of amino acid $a$ at position $r$. Model parameters were determined by 8-fold cross-validation before being applied to an independent test dataset (Lin & Hu, 2013) containing 20 protein sequences with 30 NLSs that were not examined during training.

To apply the final model to the nuclear human proteome, the test dataset was used to generate two score cutoffs: one corresponding to a precision of ~0.9 ("high-confidence NLS") and one corresponding to a recall of ~0.9 ("candidate NLS"). All 11mers present in

proteins annotated as nuclear by the Human Protein Atlas were then subject to scoring by the model. An R-markdown file with the code used to run the analysis is available on the GitHub repository.

### Time-lapse imaging of cells treated with paclitaxel

hTERT RPE-1 cells expressing Dendra2-NLS, H2B-miRFP703, and mBeRFP-NES were plated at a density of 50,000 cells per well in 2-well μm-slide chambers (ibidi; Martinsried, Germany). Twenty-four hours after plating, the cell media was replaced with media containing 0.25 nM paclitaxel. After the cell media change, the cells were imaged for 24 h with a pass time of 10 min. Imaging was performed on a Leica DMi8/Yokogawa spinning disk confocal microscope with a 20 × 0.8NA air objective at 37°C and 5% $CO_2$. Images were captured with an Andor (Belfast, United Kingdom) iXon Ultra camera using Metamorph software. Videos were cropped and adjusted for brightness and contrast using ImageJ and Photoshop.

### Visual cell sorting of cells treated with paclitaxel

RPE-1 NLS-Dendra2 × 3/H2B-miRFP/NES-mBeRFP Clone 3 cells were plated at 50,000 cells per well in a 6-well plate. After 24 h, cells were treated with paclitaxel at a final concentration of 0.25 nM. After 30 h of treatment, cells were placed on the microscope and imaged, analyzed, and activated across 2,204 sites (~75% coverage, avoided well edges) in 2 wells. At each site, Dendra2 was imaged with 1 × 1 binning; a custom nuclear segmentation pipeline that optimized detection of nuclear blebs, herniations, and other abnormalities was employed (see Appendix); Metamorph's MDA analysis was used to compute shape factors for nuclear binaries. Cells with nuclear shape factor > 0.65 were activated for 200 ms, and cells with nuclear shape factor < 0.65 were activated for 800 ms. Cells from each well were trypsinized and resuspended in DPBS supplemented with 1% BSA and 2% FBS. Using FACS, cells corresponding to 200-ms and 800-ms photoactivation were sorted using FACS (Fig EV4A) into a 1.5-ml tube containing 1 ml DPBS supplemented with 1% BSA. In Experiment 1, cells were sorted according to their nuclear phenotype (i.e. 200-ms cells in bin 1, 800-ms cells in bin 2; Appendix Fig S4A). Cells were imaged, activated, and sorted identically in Experiment 2, except that all activated cells were sorted into one bin (i.e. both 200-ms and 800-ms cells in bin 1; "unseparated paclitaxel-treated population"). For an example of the gating scheme, see Appendix Fig S3.

### Single-cell RNA sequencing of sorted, paclitaxel-treated populations

After sorting, cells were spun at 1,000 × *g* at 4°C for 5 min, and then, all but 50–100 μl of supernatant was removed. Cells were counted and subjected to 10× Single-Cell RNA sequencing v2 (10× Genomics; Pleasanton, CA; cat. no. 120236, 12037) according to the manufacturer's instructions. 10× Cell Ranger version 2.1.1 was used to process lanes corresponding to the single-cell libraries and map reads to the human reference genome build Hg19. Unique molecular identifier (UMI) cutoffs were chosen by 10× Cell Ranger software. Reads and cell numbers were normalized via downsampling by the aggregate function in 10× Cell Ranger. After normalization, cells had a median of 9,249 UMIs (Experiment 1, separated populations) or 16,932 (Experiment 2, unseparated population) per cell.

### Analysis of single-cell RNA sequencing data

Analysis of 10× CellRanger output files was done in RStudio v1.1.456 running R 3.6.0. Cell cycle scoring and annotations were performed with Seurat, as previously described (Butler *et al*, 2018). UMAP was performed with Monocle3 (Trapnell *et al*, 2014; Qiu *et al*, 2017). Mutual nearest neighbors batch correction was performed using the Batchelor package (Haghverdi *et al*, 2018) in the following order: Unseparated cells from Experiment 2 were batch corrected with morphologically normal cells from Experiment 1, and then, lobulated cells from Experiment 1 were batch corrected. An R-markdown file with the code used to run the analysis is available on the GitHub repository.

### Differentially expressed genes analysis

Mutual nearest neighbors batch correction (Haghverdi *et al*, 2018) was used to align cells from Experiment 2 (normal and lobulated cells sorted into the same tube, one 10× lane) to cells from Experiment 1 (normal and lobulated cells sorted into separate tubes, two 10× lanes). Principle components 1 through 4, which were output by the batch correction algorithm, were used to train a logistic regression model for nuclear lobulation on the cells in Experiment 1. This model was applied to Experiment 2, resulting in each cell being assigned a lobulation score, which is high in lobulated cells in Experiment 1 and low in normal cells in Experiment 1. Then, a differentially expressed gene test was performed on the cells in Experiment 2 using lobulation score, Seurat-computed G1 score, and Seurat-computed G2/M score as covariates. For a detailed discussion of this analysis, see the Appendix.

### Gene set enrichment analysis

Gene set enrichment analysis was performed using the piano package (Väremo *et al*, 2013) in R on differentially expressed genes with a $log_2$-normalized effect value (equivalent to the expected $log_2$ fold change per unit increase in lobulation score) < −0.1 and a *q*-value < 0.01. The MSigDB Hallmarks and Canonical Pathways gene sets were used (Subramanian *et al*, 2005; Liberzon *et al*, 2015).

## Data availability

The datasets and computer code produced in this study are available in the following databases:

- Bulk RNA sequencing data: Gene Expression Omnibus GSE141030 (https://www.ncbi.nlm.nih.gov/geo/query/acc.cgi?acc = GSE141 030)
- Variant sequencing data: Gene Expression Omnibus GSE141030 (https://www.ncbi.nlm.nih.gov/geo/query/acc.cgi?acc = GSE141 030)
- Single-cell RNA sequencing data: Gene Expression Omnibus GSE141030 (https://www.ncbi.nlm.nih.gov/geo/query/acc.cgi? acc = GSE141030)
- Code for all analysis and figure generation: GitHub (https:// github.com/FowlerLab/vcs_2019)
- Flow cytometry data: GitHub (https://github.com/FowlerLab/vc s_2019)
- Imaging datasets are available upon request.

**Expanded View** for this article is available online.

## Acknowledgements

Kate Sitko, Gabriel Boyle, and Kathleen Abadie contributed to the development of Visual Cell Sorting and helped refine its capabilities. Jose McFaline provided advice for the interpretation and analysis of single-cell RNA sequencing results. Sri Kosuri provided valuable insight and encouragement. Stanley Fields and Brian Beliveau provided helpful comments on the manuscript. This work was funded by the NIH (F30CA236335-01 to N.H., R01GM109110 and RM1HG010461 to D.M.F., R35GM124766-03 and P30CA015704 to E.M.H, DP2HD088158 to C.T., R01HL120948 to R.J.M Jr.), the W. M. Keck Foundation (C.T.), the NSF (DGE-1258485 to S.S) and The Paul G. Allen Frontiers Group (C.T.).

## Author contributions

NH and DMF conceived of Visual Cell Sorting. NH and AC carried out Visual Cell Sorting experiments. SS, JJS, and DJ prepared next-generation sequencing libraries. ZK validated NLS variant scores. SP cloned constructs and helped with preliminary experiments. EMH guided the paclitaxel experiment. WT and RJM Jr. created the U-2 OS AAVS-LP Clone 11 cells. EMH and HH provided hTERT RPE-1 cells and performed time-lapse imaging. SS and CT provided valuable insight for the analysis of single-cell RNA sequencing results. NH and DMF wrote the manuscript. All authors commented on the manuscript.

## Conflict of interest

The authors declare that they have no conflict of interest.

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
