## [Review Process File · Molecular Systems Biology]

High-throughput, Microscope-based Sorting to Dissect Cellular Heterogeneity

Nicholas Hasle, Tony Cooke, Sanjay Srivatsan, Heather Huang, Jason Stephany, Zachary Krieger, Dana Jackson, Weiliang Tang, Sriram Pendyala, Raymond Monnat, Cole Trapnell, Emily Hatch, and Douglas Fowler

DOI: [10.15252/msb.20209442](https://doi.org/10.15252/msb.20209442)

Corresponding author(s): Douglas Fowler (dfowler@uw.edu)

Review Timeline:

Submission Date:	7th Jan 20
Editorial Decision:	18th Feb 20
Revision Received:	3rd Apr 20
Editorial Decision:	13th Apr 20
Revision Received:	23rd Apr 20
Accepted:	29th Apr 20

Editor: Jingyi Hou

Transaction Report:

18th Feb 2020

Manuscript Number: MSB-20-9442

Title: High-throughput, Microscope-based Sorting to Dissect Cellular Heterogeneity

Author: Nicholas Hasle

Tony Cooke

Sanjay Srivatsan

Heather Huang

Jason Stephany

Zachary Krieger

Dana Jackson

Weiliang Tang

Sriram Pendyala

Raymond Monnat

Cole Trapnell

Emily Hatch

Douglas Fowler

Thank you for submitting your work to Molecular Systems Biology. We have now heard back from two of the three reviewers who agreed to evaluate your manuscript. Since their recommendations are rather similar, I prefer to make a decision now rather than further delaying the process. As you will see below, the reviewers think that the presented method and findings seem interesting. They raise however a series of concerns, which we would kindly ask you to convincingly address in a revision. The recommendations provided by the reviewers are very clear and there is therefore no need to repeat their comments. Please feel free to contact me in case you would like to discuss in further detail any of the issues raised by the reviewers.

On a more editorial level, please do the following:

- Please provide a .docx formatted version of the manuscript text (including legends for main figures, EV figures and tables). Please make sure that the changes are highlighted to be clearly visible.
- Please provide individual production quality figure files as .eps, .tif, .jpg (one file per figure).
- Please provide a .docx formatted letter INCLUDING the reviewers' reports and your detailed point-by-point responses to their comments. As part of the EMBO Press transparent editorial process, the point-by-point response is part of the Review Process File (RPF), which will be published alongside your paper.
- Please note that all corresponding authors are required to supply an ORCID ID for their name upon submission of a revised manuscript.
- We have replaced Supplementary Information by the Expanded View (EV format). In this case, all additional Figures can be provided as EV Figures. Please provide the images as individual files and

include the EV Figure legends in the main text together with the main Figure legends. For detailed instructions regarding Expanded View please refer to our Author Guidelines:

<https://www.embopress.org/page/journal/17444292/authorguide#expandedview>.

- The tables currently provided as Supplementary Tables 1-8 should be provided (and called out in the text) as Datasets EV1-EV8. Please provide each of them as an .xls files and include a brief description of the Dataset in a separate tab.

-- Before submitting your revision, primary datasets (and computer code, where appropriate) produced in this study need to be deposited in an appropriate public database (see <https://www.embopress.org/page/journal/17444292/authorguide#dataavailability>). - Dataset #1
- Dataset #2>

The accession numbers and database should be listed in a formal "Data Availability " section (placed after Materials & Method) that follows the model below (see also <https://www.embopress.org/page/journal/17444292/authorguide#dataavailability>). Please note that the Data Availability Section is restricted to new primary data that are part of this study.

Data availability

- We would encourage you to include the source data for figure panels that show essential quantitative information. Additional information on source data and instruction on how to label the files are available at < <https://www.embopress.org/page/journal/17444292/authorguide#sourcedata> >.

- We would kindly ask you to use 'Structured Methods', our new Materials and Methods format, which is mandatory for Method papers. According to this format, the Material and Methods section should include a Reagents and Tools Table (listing key reagents, experimental models, software and relevant equipment and including their sources and relevant identifiers) followed by a Methods and Protocols section in which we encourage the authors to describe their methods using a step-by-step protocol format with bullet points, to facilitate the adoption of the methodologies across labs. More information on how to adhere to this format as well as downloadable templates (.doc or .xls) for the Reagents and Tools Table can be found in our author guidelines: < <https://www.embopress.org/page/journal/17444292/authorguide#researcharticleguide> >. An example of a Method paper with Structured Methods can be found here: .

- Please provide a "standfirst text" summarizing the study in one or two sentences (approximately 250 characters, including space), three to four "bullet points" highlighting the main findings and a "synopsis image" (550px width and max 400px height, jpeg format) to highlight the paper on our

homepage.

- When you resubmit your manuscript, please download our CHECKLIST (http://embopress.org/sites/default/files/Resources/EP_Author_Checklist.xls) and include the completed form in your submission. *Please note* that the Author Checklist will be published alongside the paper as part of the transparent process <http://msb.embopress.org/authorguide#transparentprocess>.

If you feel you can satisfactorily deal with these points and those listed by the referees, you may wish to submit a revised version of your manuscript. Please attach a covering letter giving details of the way in which you have handled each of the points raised by the referees. A revised manuscript will be once again subject to review and you probably understand that we can give you no guarantee at this stage that the eventual outcome will be favorable.

If you do choose to resubmit, please click on the link below to submit the revision online *within 90 days*.

Link Not Available

IMPORTANT: When you send your revision, we will require the following items:

1. the manuscript text in LaTeX, RTF or MS Word format
2. a letter with a detailed description of the changes made in response to the referees. Please specify clearly the exact places in the text (pages and paragraphs) where each change has been made in response to each specific comment given
3. three to four 'bullet points' highlighting the main findings of your study
4. a short 'blurb' text summarizing in two sentences the study (max. 250 characters)
5. a 'thumbnail image' (550px width and max 400px height, Illustrator, PowerPoint or jpeg format), which can be used as 'visual title' for the synopsis section of your paper.
6. Please include an author contributions statement after the Acknowledgements section (see <https://www.embopress.org/page/journal/17444292/authorguide>)
7. Please complete the CHECKLIST available at (<http://bit.ly/EMBOPressAuthorChecklist>). Please note that the Author Checklist will be published alongside the paper as part of the transparent process (<https://www.embopress.org/page/journal/17444292/authorguide#transparentprocess>).
8. Please note that corresponding authors are required to supply an ORCID ID for their name upon submission of a revised manuscript (EMBO Press signed a joint statement to encourage ORCID adoption). (<https://www.embopress.org/page/journal/17444292/authorguide#editorialprocess>)

Currently, our records indicate that there is no ORCID associated with your account.

Please click the link below to provide an ORCID:

Link Not Available

The system will prompt you to fill in your funding and payment information. This will allow Wiley to send you a quote for the article processing charge (APC) in case of acceptance. This quote takes into account any reduction or fee waivers that you may be eligible for. Authors do not need to pay any fees before their manuscript is accepted and transferred to the publisher.

*** PLEASE NOTE *** As part of the EMBO Press transparent editorial process initiative (see our Editorial at <http://dx.doi.org/10.1038/msb.2010.72>), Molecular Systems Biology publishes online a Review Process File with each accepted manuscripts. This file will be published in conjunction with your paper and will include the anonymous referee reports, your point-by-point response and all pertinent correspondence relating to the manuscript. If you do NOT want this File to be published, please inform the editorial office at msb@embo.org within 14 days upon receipt of the present letter.

Reviewer #2:

Summary

The authors present a very clever method, termed Visual Cell Sorting, to enable pooled genetic perturbation screens (as well as other studies of cell heterogeneity); which has been a huge unmet need. They photo-activate cells based on their visual appearance by microscopy, which then allows these cells to be physically sorted then sequenced or otherwise studied.

General remarks

Other techniques exist to do this, such as Pooled Optical Profiling from the Blainey lab at MIT and others reviewed in the introduction, but Visual Cell Sorting offers some major advantages (namely, reagent cost, throughput, and simplicity of instrumentation required) and will be more appropriate in many cases, such as when the number of classes of cells to be defined is relatively small (up to 4). The method is novel to my knowledge and seems well worth consideration for correlating genomic/transcriptomic information with morphology. The analysis and photo activation control pipeline is automated so a phenotype can be pre-defined and then the system marks cells automatically, enabling a decent volume of cells to be assessed and later sorted (~650,000 demonstrated here). Controls are good, and include testing for phototoxicity.

The applications demonstrated produced beautiful, rich data: computing a quantitative score for nuclear localization for ~200 single missense variants in an NLS (identifying a novel powerful synthetic NLS), and deriving RNA-seq profiles for cells responding differentially to a chemotherapy drug. Limitations and considerations are nicely described in the Discussion section. The figures are nicely organized and clear, as is the writing. Overall, this is very strong work and a tool likely to be useful to a wide range of laboratories, and readily adopted.

Major points

- the editors should ensure that the code is provided as claimed on Github upon publication, and that it includes the MetaMorph journals as well as other code claimed:

https://github.com/FowlerLab/vcs_2019.git

Minor points

- some figure panels have light gray boxes around them which are distracting
- in the supplement, "In all experiments, nuclei touching the image border were removed." Is a bit confusing. This likely means they are ignored in the image analysis but does this mean those cells do not get photo activated? What proportion of cells are edge cases and thus impossible to be assessed properly?

Reviewer #3:

A simple platform for imaging-based sorting would greatly expand the range of phenotypes amenable to high-throughput screening. Here Hasle et al. describe an approach that uses a fluorescence microscope to label cells exhibiting different phenotypes (up to 4 phenotypic bins) by photoconversion of a reporter fluorescent protein, followed by FACS to physically separate differentially labeled populations. The approach makes use of commercially available hardware and software, and thus should be of great interest to many laboratories. The authors showcase their approach in two applications: saturation mutagenesis of a nuclear localization sequence and transcriptomics of single cells treated with the microtubule stabilizer paclitaxel. Overall I find the experiments are well performed and the manuscript is clearly written, with detailed descriptions of the methodology.

Major points

1. In the first application, Hasle et al. show how their approach can be used to perform saturation mutagenesis experiments with SV40 NLS as an example. Using the resulting data set, the authors trained an NLS predictor and used it to identify potential NLS sequences in human nuclear proteins. This section would be greatly strengthened if the authors could apply their imaging-based approach to actually test the predicted NLS sequences for functionality. In the absence of such testing, at the very least it would be interesting to know how their predictor performs on human proteins that are not nuclear.
2. In their second application, Hasle et al. examined cells treated with paclitaxel. Based on single-cell RNA seq, the authors conclude that, among cells exposed to paclitaxel, those exhibiting normal nuclear morphology mount a biosynthetic and proteostatic response to the treatment. The main issue here is that the authors do not perform RNA seq of untreated cells. Without this control it is unclear how the authors can arrive at their conclusion. How can they distinguish between (i) "normal nuclei" cells mounting a response vs. (ii) cell-to-cell differences in gene expression before drug treatment simply being revealed by the treatment? The presented data is consistent with the possibility that cells that happened to express less CCT, TUBB4B, etc before treatment are more sensitive to paclitaxel treatment and thus are the ones acquiring the "lobulated nuclei" phenotype.
3. The authors do a fair job of contrasting their approach with existing methods. Nevertheless, it would be important to have a bit more discussion of its potential limitations, for instance its application in faster dividing cells such as yeast or e.coli and if/how the authors envision their approach could be used to split populations in more than 4 phenotypic bins.

Minor points

1. It would be useful to see in the paper example fields-of-view before and after photoconversion in both green and red channels. Especially for the case of 4-bin activation.
2. It's important that the authors provide the tables as individual supplementary files (csv, xlsx, ...) instead of embedding them all in the pdf. As such, the tables are unusable.
3. When appropriate, to improve clarity the authors should refer to specific panels in the supplementary figures when calling out supplementary figures throughout the manuscript.
4. If I am interpreting the mutagenesis results in Fig. 2d and 3b correctly, then the SuperNLS sequence should be EPPRKKRKIGI, not EPPKRKKRKII (as indicated in Fig. 3c) or PPRKKRKI (second paragraph of the Discussion).
5. Legend to figure S2. Panel C - it is not clear which channel is shown: miRFP or one of the Dendra2 channels. Panel D - it is not clear from the legend what "scores" are shown here.
6. Can the authors clarify in the methods section why they assayed 209 single residue NLS mutants using 346 nucleotide variants?

Reviewer #2:

Summary

The authors present a very clever method, termed Visual Cell Sorting, to enable pooled genetic perturbation screens (as well as other studies of cell heterogeneity); which has been a huge unmet need. They photo-activate cells based on their visual appearance by microscopy, which then allows these cells to be physically sorted then sequenced or otherwise studied.

General remarks

Other techniques exist to do this, such as Pooled Optical Profiling from the Blainey lab at MIT and others reviewed in the introduction, but Visual Cell Sorting offers some major advantages (namely, reagent cost, throughput, and simplicity of instrumentation required) and will be more appropriate in many cases, such as when the number of classes of cells to be defined is relatively small (up to 4). The method is novel to my knowledge and seems well worth consideration for correlating genomic/transcriptomic information with morphology. The analysis and photo activation control pipeline is automated so a phenotype can be pre-defined and then the system marks cells automatically, enabling a decent volume of cells to be assessed and later sorted (~650,000 demonstrated here). Controls are good, and include testing for phototoxicity.

The applications demonstrated produced beautiful, rich data: computing a quantitative score for nuclear localization for ~200 single missense variants in an NLS (identifying a novel powerful synthetic NLS), and deriving RNA-seq profiles for cells responding differentially to a chemotherapy drug. Limitations and considerations are nicely described in the Discussion section. The figures are nicely organized and clear, as is the writing. Overall, this is very strong work and a tool likely to be useful to a wide range of laboratories, and readily adopted.

Thank you! We are excited about the method, too.

Major points

1. The editors should ensure that the code is provided as claimed on Github upon publication, and that it includes the MetaMorph journals as well as other code claimed:

https://github.com/FowlerLab/vcs_2019.git

The GitHub repository was private at the time of submission. We have now made it public, and it includes all code and MetaMorph journals referenced in the paper. Additionally, it contains the source data that is not posted in GEO.

Minor points

1. Some figure panels have light gray boxes around them which are distracting.

We did not intend for any figure panels to be set in gray boxes. Perhaps there was an issue with the PDF conversion. If these boxes still exist, please tell us which figure/panels are affected and we will try to fix the issue.

2. In the supplement, "In all experiments, nuclei touching the image border were removed." Is a bit confusing. This likely means they are ignored in the image analysis but does this mean those cells do not get photo activated? What proportion of cells are edge cases and thus impossible to be assessed properly?

To address this question, we examined 10 randomly chosen images from replicate 2, technical replicate 1 of the NLS experiment. We found that approximately 7% of cells are on image borders and thus would have been removed from the analysis. These cells would end up being un-activated. We acknowledge that this contaminates our lowest bin for the NLS experiments and reduces our power somewhat. In the paclitaxel experiments, because only activated cells were sorted, these “missed” cells are not an issue. We feel that discarding cells at the edges is the best we can do, because analyzing and activating nuclei touching the image border would lead to inaccurate phenotyping (e.g. for nuclear shape) and partial activation.

To make this issue clear to the reader we have added the following sentence to the supplement “Removed nuclei are not photoactivated, and thus end up in the unactivated bin. Using the imaging conditions and cell type presented here, ~7% of cells are on image borders and thus are removed.”

Reviewer #3:

Summary

A simple platform for imaging-based sorting would greatly expand the range of phenotypes amenable to high-throughput screening. Here Hasle et al. describe an approach that uses a fluorescence microscope to label cells exhibiting different phenotypes (up to 4 phenotypic bins) by photoconversion of a reporter fluorescent protein, followed by FACS to physically separate differentially labeled populations. The approach makes use of commercially available hardware and software, and thus should be of great interest to many laboratories. The authors showcase their approach in two applications: saturation mutagenesis of a nuclear localization sequence and transcriptomics of single cells treated with the microtubule stabilizer paclitaxel. Overall I find the experiments are well performed and the manuscript is clearly written, with detailed descriptions of the methodology.

Thank you for this positive evaluation of our work.

Major points

1. In the first application, Hasle et al. show how their approach can be used to perform saturation mutagenesis experiments with SV40 NLS as an example. Using the resulting data set, the authors trained an NLS predictor and used it to identify potential NLS sequences in human nuclear proteins. This section would be greatly strengthened if the authors could apply their imaging-based approach to actually test the predicted NLS sequences for functionality. In the absence of such testing, at the very least it would be interesting to know how their predictor performs on human proteins that are not nuclear.

We agree that the results of our predictor would be strengthened by validating some of the NLS sequences. However, using our reporter to conduct these experiments comes with substantial caveats, because validating a sequence out of its native context could profoundly alter NLS function. For example, removing an NLS from its native context could eliminate nearby sequences that control NLS conformation or bind the minor groove of importin alpha. Furthermore, our reporter requires exceptionally strong importin alpha binding for nuclear import because the size of tetrameric NLS-CMPK-miRFP protein is over 360 kDa, ~6-fold larger than most proteins. Thus, the reporter is not a physiologic cargo size for the NLS's in smaller proteins. In order to fairly evaluate candidate predicted NLS's, we would have to clone and evaluate a significant number of different proteins, which we feel is beyond the scope of this manuscript.

To assess how the predictor performs on non-nuclear proteins, we compared the best-scoring 11mer for proteins observed to be only nuclear to those observed to be only cytoplasmic in the Human Protein Atlas (N = 3,925 proteins). We see a clear difference between the two groups of proteins: Twenty-eight percent of nucleus-only proteins have 11mers with scores above our high-confidence NLS score cutoff, (score > 0.0243), whereas only 11% of cytoplasm-only proteins have such 11mers (sensitivity = 0.28, specificity = 0.89). Using an even more stringent cutoff (score > 0.0275), we obtain a sensitivity of 0.09 and a specificity of 0.99 for identifying nucleus-only proteins.

We note that there are multiple explanations for “cytoplasmic” proteins with a high-scoring 11-mer: (a) the protein contains a functional NLS that becomes accessible to the nuclear import machinery after a signaling event (e.g. NFkB; see

PMIDs 1340770 and 1547506), which may not be active under the Human Protein Atlas imaging conditions or (b) the protein contains a nuclear export signal that overwhelms an otherwise functional NLS (e.g. viral Rev proteins). The “nuclear” proteins with low NLS-scoring 11mers might arise from (a) an interaction partner with a functional NLS that brings them into the nucleus or (b) a non-SV40 type NLS.

We have updated Supplementary Figure 3B and 3C, and included the following text discussing these points in the manuscript:

Page 9, paragraph 2:

To substantiate that these represent bona-fide NLS sequences, we compared the top-scoring 11-mers in exclusively nuclear proteins to those in exclusively cytoplasmic proteins (Figure EV3B, C). As expected, nuclear proteins had higher top-scoring 11mer sequences than cytoplasmic proteins (Wilcoxon rank sum p value < 10⁻¹⁶). Twenty-eight percent of the nucleus-only proteins contained an 11-mer with an NLS score higher than our high-confidence cutoff; only 11% of cytoplasmic proteins contained such a sequence. These results are consistent with our predictor identifying monopartite, SV40-like NLS's in the human proteome.

Page 12, final paragraph:

We then used the variant scores to make an accurate, amino acid preference-based predictor of NLS function, which we applied to the human nuclear proteome and validated by comparing the top-scoring sequences between cytoplasmic and nuclear proteins. Interestingly, some cytoplasmic proteins contain putative NLS's, which could be explained by an NLS that becomes accessible to the nuclear import machinery after a signaling event⁴⁴ or a nuclear export signal located on the same protein that overwhelms an otherwise functional NLS⁴⁵. Nuclear proteins without high-scoring sequences may harbor a non-SV40 type NLS or have an interaction partner with a functional NLS enables co-import into the nucleus.

2. In their second application, Hasle et al. examined cells treated with paclitaxel. Based on single-cell RNA seq, the authors conclude that, among cells exposed to paclitaxel, those exhibiting normal nuclear morphology mount a biosynthetic and proteostatic response to the treatment. The main issue here is that the authors do not perform RNA seq of untreated cells. Without this control it is unclear how the authors can arrive at their conclusion. How can they distinguish between (i) "normal nuclei" cells mounting a response vs. (ii) cell-to-cell differences in gene expression before drug treatment simply being revealed by the treatment? The presented data is consistent with the possibility that cells that happened to express less CCT, TUBB4B, etc before treatment are more sensitive to paclitaxel treatment and thus are the ones acquiring the "lobulated nuclei" phenotype.

This is an excellent question; whether the observed gene expression differences between normal and lobulated cells pre-exist before drug treatment, or are induced stochastically after the drug is added, cannot be answered using our dataset. However, we do not believe that performing Visual Cell Sorting and then sequencing DMSO-treated cells would adequately answer this question. Rather, such an experiment would tell us whether the association between visual

phenotype and RNA expression profiles is the same in DMSO and paclitaxel-treated cells.

To answer the question of pre-existing vs. paclitaxel-induced resistant states, one could (a) perform live-cell tracking of cells that express fluorescent reporters of the genes (e.g. TUBB4B) and pathways (e.g. c-Myc) we identify before and for 48 hours after paclitaxel is added; or (b) use cellular barcoding methods (e.g. CellTagging, see PMID 31072405) together with Visual Cell Sorting to examine whether cell sisters tend to reach the same phenotype and RNA expression profiles. As exciting as these experiments are, we feel that they are outside the scope of this manuscript and are better addressed in another publication.

Finally, we note that the normal-appearing cells could be analogous to the “pre-resistant” melanoma cells that were discovered by Shaffer and colleagues (see PMID 28607484). Rather than possess resistance-conferring mutations, these cells exhibit stochastically activated gene expression programs that assist with cell survival after drug treatment.

We have updated the manuscript to remove language asserting that cells are “responding” to paclitaxel treatment:

Page 12, 1st paragraph, last sentence:

Together, these results suggest that the morphologically normal, paclitaxel-treated cells exhibit ~~mounted~~ a biosynthetic and proteostatic gene expression program ~~response to drug treatment~~, with remarkable similarities to the gene expression profiles observed in paclitaxel resistant cell lines and cancers.

Furthermore, we have added language to the discussion regarding the points raised above:

Page 13, second paragraph:

To our surprise, cells that resist the effect of paclitaxel on nuclear morphology appear to be ~~adapting to counteracting~~ the drug’s effects at the molecular level using with a gene expression program similar to paclitaxel-resistant cancers. This phenomenon, whereby a subset of clonal cells resists the effects of drug treatment with a protective gene expression program, is reminiscent of the “pre-resistance” reported in primary melanoma cells⁴⁶. However, the experiment we conducted cannot determine whether the gene expression program pre-exists in the population or is stochastically induced upon paclitaxel addition. To answer this question, live-cell microscopy or cell barcoding could be used to determine whether pre-treatment levels of the genes expressed highly in morphologically normal cells (e.g. TUBB4B, c-Myc targets) leads to morphologic responses and survival after paclitaxel treatment.

3. The authors do a fair job of contrasting their approach with existing methods. Nevertheless, it would be important to have a bit more discussion of its potential limitations, for instance its application in faster dividing cells such as yeast or e.coli and if/how the authors envision their approach could be used to split populations in more than 4 phenotypic bins.

These are important points. We now explain in the discussion section that the number of phenotypic bins, four, is limited by current fluorescence activated cell

sorting technology. We also discuss how the signal decay timescale makes it challenging to study transient phenotypes (e.g. cell-cycle dependent phenotypes) and may make it more challenging to study rapidly dividing cells, such as yeast or bacterial cells.

The specific language is copied below:

Page 14, final paragraph, 5th line:

Another limitation is that, unlike morphological profiling approaches⁵¹, Visual Cell Sorting requires a pre-defined phenotype of interest and is limited by current FACS hardware to 4 phenotypic bins.

Page 14, final paragraph, 7th line:

The several hours required to execute a Visual Cell Sorting experiment makes it challenging to study transient phenotypes (e.g. cell-cycle dependent phenotypes). Furthermore, decay of photoactivated Dendra 2 may be more pronounced in rapidly dividing bacteria or yeast as activated Dendra2 is diluted by cell division.

Minor points

1. It would be useful to see in the paper example fields-of-view before and after photoconversion in both green and red channels. Especially for the case of 4-bin activation.

We have included in Supplementary Figure 1A images of cells before and after photoactivation.

2. It's important that the authors provide the tables as individual supplementary files (csv, xlsx, ...) instead of embedding them all in the pdf. As such, the tables are unusable.

As a result of the PDF creation process for review, tables were converted to PDFs. We will ensure that MSB provides tables in the .xls formats we submitted for publication.

3. When appropriate, to improve clarity the authors should refer to specific panels in the supplementary figures when calling out supplementary figures throughout the manuscript.

The manuscript has been updated to include specific panels in references to supplementary figures.

4. If I am interpreting the mutagenesis results in Fig. 2d and 3b correctly, then the SuperNLS sequence should be EPPRKKRKIGI, not EPPKRKKRKII (as indicated in Fig. 3c) or PPRKKRKI (second paragraph of the Discussion).

Thank you for noticing this important error. The manuscript and figure have been updated.

5. Legend to figure S2. Panel C - it is not clear which channel is shown: miRFP or one of the Dendra2 channels. Panel D - it is not clear from the legend what "scores" are shown here.

We have specified the channel imaged for Panel C (miRFP) and added axis titles to Panel D ("Localization Scores Replicate A/B").

6. Can the authors clarify in the methods section why they assayed 209 single residue NLS mutants using 346 nucleotide variants?

To address this question, we added the following sentence to the methods section (page 59, 1st paragraph, 2nd sentence):

The final library contained 346 NNK nucleotide variants which, due to codon degeneracy in the genetic code, encode for 209 single amino acid variants.

13th Apr 2020

Manuscript Number: MSB-20-9442R

Title: High-throughput, Microscope-based Sorting to Dissect Cellular Heterogeneity

Author: Nicholas Hasle

Tony Cooke

Sanjay Srivatsan

Heather Huang

Jason Stephany

Zachary Krieger

Dana Jackson

Weiliang Tang

Sriram Pendyala

Raymond Monnat

Cole Trapnell

Emily Hatch

Douglas Fowler

Thank you for sending us your revised manuscript. We have now heard back from the two reviewers who were asked to evaluate your study. As you will see the reviewers are satisfied with the modifications made and think that the study is now suitable for publication.

Before we can formally accept your manuscript, we would ask you to address a few remaining editorial issues listed below:

1. please provide 5 keywords and incorporate them in the main text.
2. Please check the figure and dataset callouts in the main article and make sure that they are correctly called for. Currently, Dataset EV 4 & 5 are not called out, please fix. On Page 11, Table EV4 is called out, while only 3 EV tables are provided, please fix.
3. Please list 10 co-authors of a paper before to add et al. in the reference list. Citations should be listed in alphabetical order, with the authors' surnames and initials inverted. More information can be found at <https://www.embopress.org/page/journal/17444292/authorguide#referencesformat>
4. Appendix file: Figures are not correctly labeled. They should be Appendix figure S1, S2 & S3 and the Legends should be clearly marked. Please add a Table of Content on the 1st page and incorporate in this single pdf file the tables as well.
The last page of Appendix (the one that is uploaded as a pdf) seems missing, please make sure that it's complete.
Also, please remove the appendix section from the main manuscript.
5. Please use 'Structured Methods', our new Materials and Methods format, which is mandatory for Method papers. According to this format, the Material and Methods section should include a Reagents and Tools Table (listing key reagents, experimental models, software and relevant

equipment and including their sources and relevant identifiers) followed by a Methods and Protocols section in which we encourage the authors to describe their methods using a step-by-step protocol format with bullet points, to facilitate the adoption of the methodologies across labs. More information on how to adhere to this format as well as downloadable templates (.doc or .xls) for the Reagents and Tools Table can be found in our author guidelines: <
<https://www.embopress.org/page/journal/17444292/authorguide#researcharticleguide>>. An example of a Method paper with Structured Methods can be found here: .

6. Synopsis image: The current synopsis image appears somewhat too simplified. A synopsis image is supposed to provide a visual summary of the paper. Please add some simple text to the synopsis image (550px width and max 400px height, jpeg format), to make it more accessible to the reader.

More examples of synopsis image can be found here:
<https://www.embopress.org/doi/10.15252/msb.20198875>
<https://www.embopress.org/doi/10.15252/msb.20199170>
<https://www.embopress.org/doi/10.15252/msb.20199195>

7. I have slightly modified the synopsis text. Could you let me know if you would like to introduce further modifications?

This study describes an imaging-based approach for pooled genetic screening that uses high-throughput photoactivation of visually defined cell subpopulations, followed by FACS to separate differentially labeled populations.

- Expression of the photoactivatable fluorescent protein Dendra2 permits selective, irreversible, and high-throughput labeling of cells exhibiting different visual phenotypes, These labeled cell subpopulations can be sorted and thus subject to diverse downstream genomics assays.
- Photoactivation using a digital micromirror device affixed to a 405 nm laser is accurate, non-toxic, and can be tuned to produce four discrete levels of fluorescence.
- Human cells expressing sequence variant libraries can be sorted according to a visual phenotype followed by sequencing, which provides sequence-function maps for phenotypes such as protein subcellular localization.
- Cell populations that respond in a visually heterogeneous fashion to drug treatment can be sorted and subject to transcriptomic analyses, revealing the molecular states associated with complex drug responses.

Please submit your revised manuscript within two weeks. I look forward to seeing a revised version of your manuscript.

When you resubmit your manuscript, please download our CHECKLIST (<http://bit.ly/EMBOPressAuthorChecklist>) and include the completed form in your submission. *Please note* that the Author Checklist will be published alongside the paper as part of the transparent process (<https://www.embopress.org/page/journal/17444292/authorguide#transparentprocess>)

Click on the link below to submit your revised paper.

Link Not Available

If you do choose to resubmit, please click on the link below to submit the revision online before 13th May 2020.

Link Not Available

IMPORTANT: When you send your revision, we will require the following items:

1. the manuscript text in LaTeX, RTF or MS Word format
2. a letter with a detailed description of the changes made in response to the referees. Please specify clearly the exact places in the text (pages and paragraphs) where each change has been made in response to each specific comment given
3. three to four 'bullet points' highlighting the main findings of your study
4. a short 'blurb' text summarizing in two sentences the study (max. 250 characters)
5. a 'thumbnail image' (550px width and max 400px height, Illustrator, PowerPoint or jpeg format), which can be used as 'visual title' for the synopsis section of your paper.
6. Please include an author contributions statement after the Acknowledgements section (see <https://www.embopress.org/page/journal/17444292/authorguide#manuscriptpreparation>)
7. Please complete the CHECKLIST available at (<http://bit.ly/EMBOPressAuthorChecklist>). Please note that the Author Checklist will be published alongside the paper as part of the transparent process (<https://www.embopress.org/page/journal/17444292/authorguide#transparentprocess>).
8. Please note that corresponding authors are required to supply an ORCID ID for their name upon submission of a revised manuscript (EMBO Press signed a joint statement to encourage ORCID adoption) (<https://www.embopress.org/page/journal/17444292/authorguide#editorialprocess>).

Currently, our records indicate that the ORCID for your account is 0000-0001-7614-1713.

Link Not Available

The system will prompt you to fill in your funding and payment information. This will allow Wiley to send you a quote for the article processing charge (APC) in case of acceptance. This quote takes into account any reduction or fee waivers that you may be eligible for. Authors do not need to pay any fees before their manuscript is accepted and transferred to the publisher.

As a matter of course, please make sure that you have correctly followed the instructions for

authors as given on the submission website.

*** PLEASE NOTE *** As part of the EMBO Press transparent editorial process initiative (see our Editorial at <http://dx.doi.org/10.1038/msb.2010.72> , Molecular Systems Biology will publish online a Review Process File to accompany accepted manuscripts. When preparing your letter of response, please be aware that in the event of acceptance, your cover letter/point-by-point document will be included as part of this File, which will be available to the scientific community. More information about this initiative is available in our Instructions to Authors. If you have any questions about this initiative, please contact the editorial office (msb@embo.org).

Reviewer #2:

The authors addressed my concerns.

Reviewer #3:

The authors have addressed all point I raised. I have no further comments or suggestions. It is a cool method and a nice paper.

Corresponding Author Name: Douglas M. Fowler

Manuscript Number: MSB-20-9442